# Fine-scale population structure and demographic history of British Pakistanis

Elena Arciero [1✉], Sufyan A. Dogra [2,10], Daniel S. Malawsky[1,10], Massimo Mezzavilla [3,10], Theofanis Tsismentzoglou [4,5], Qin Qin Huang [1], Karen A. Hunt[6], Dan Mason [2], Saghira Malik Sharif[7], David A. van Heel [6], Eamonn Sheridan[5], John Wright[2], Neil Small [8], Shai Carmi[9], Mark M. Iles [4,5,11] & Hilary C. Martin [1,11✉]

Previous genetic and public health research in the Pakistani population has focused on the role of consanguinity in increasing recessive disease risk, but little is known about its recent population history or the effects of endogamy. Here, we investigate fine-scale population structure, history and consanguinity patterns using genotype chip data from 2,200 British Pakistanis. We reveal strong recent population structure driven by the biraderi social stratification system. We find that all subgroups have had low recent effective population sizes (N$_e$), with some showing a decrease 15–20 generations ago that has resulted in extensive identity-by-descent sharing and homozygosity, increasing the risk of recessive disorders. Our results from two orthogonal methods (one using machine learning and the other coalescent-based) suggest that the detailed reporting of parental relatedness for mothers in the cohort under-represents the true levels of consanguinity. These results demonstrate the impact of cultural practices on population structure and genomic diversity in Pakistanis, and have important implications for medical genetic studies.

[1] Wellcome Sanger Institute, Wellcome Genome Campus, Hinxton, UK. [2] Bradford Institute for Health Research, Bradford Teaching Hospitals NHS Foundation Trust, Bradford, UK. [3] Department of Medical Sciences, University of Trieste, Trieste, Italy. [4] Leeds Institute for Data Analytics, University of Leeds, Leeds, UK. [5] Leeds Institute of Medical Research, University of Leeds, Leeds, UK. [6] Blizard Institute, Barts and The London School of Medicine and Dentistry, Queen Mary University of London, London, UK. [7] Yorkshire Regional Genetics Service, Leeds Teaching Hospitals NHS Trust, Leeds, UK. [8] Faculty of Health Studies, University of Bradford, Richmond Road, Bradford, UK. [9] Braun School of Public Health and Community Medicine, The Hebrew University of Jerusalem, Jerusalem, Israel. [10] These authors contributed equally: Sufyan A. Dogra, Daniel S. Malawsky, Massimo Mezzavilla. [11] These authors jointly supervised this work: Mark M. Iles, Hilary C. Martin. ✉email: ea6@sanger.ac.uk; hcm@sanger.ac.uk

Estimates suggest that around 10% of the world's population are offspring of closely related parents, mostly in the north and sub-Saharan Africa, the Middle East, and west, central, and south Asia[1]. In these regions, there is frequently co-occurring endogamy (marriage within clans)[2]. Exploring the impact of consanguinity, endogamy, and population structure on genetic variation within these populations is important for quantifying their relative effects on risks of genetic diseases. Furthermore, a proper understanding of the demographic history of a population is critical for designing robust and effective medical genetic analyses, as has been recently highlighted in work showing the impact of population structure on polygenic scores[3,4]. Here, we investigate population structure, history, and consanguinity patterns in individuals with Pakistani ancestry living in the United Kingdom (UK).

British Pakistanis are one of the largest and most socio-economically disadvantaged ethnic minorities in the UK, with a population size of 1.17 million in the 2011 census[5,6]. The most substantial wave of migration from Pakistan to the UK occurred in the 1950s/1960s, after the partition of British India[7,8]. British Pakistanis have rates of type 2 diabetes and heart disease that are two to four times higher than the white British population[9–11], as well as an increased risk of congenital anomalies due to the prevalence of consanguinity[12]. These factors, combined with the drive to increase the number of genetic studies on people with non-European ancestry[13], have spurred the creation of several cohort studies whose aims include exploring the environmental and genetic contributions to various phenotypes in Pakistani-ancestry individuals and the impact of homozygous gene knockouts[14]. These include Genes & Health[15] and Born in Bradford (BiB)[16] (in the UK), and the Pakistan Risk of Myocardial Infarction Study[17].

Most modern South Asians are a mixture of two ancestral populations: the Ancestral Northern Indian (ANI) and Ancestral Southern Indian (ASI) components[18–20]. North-west Indians and Pakistanis have a greater proportion of the ANI component[18,19]. Several major studies of human genetic diversity, such as the Human Genome Diversity Project (HGDP)[21], 1000 Genomes[22], and GenomeAsia[23], have highlighted differences between multiple ethnic groups within Pakistan, as well as the elevated levels of homozygosity due to consanguinity[24]. These previous studies have focused mostly on population structure on a macro-scale, with relatively limited sample sizes per population, and they lacked information on finer-scale groupings within each of these populations.

Most of the ethnic and tribal groups in Pakistan follow a patrilineal kinship system based on the biraderi (brotherhood) that shares its historical roots with the better-studied Indian caste system. The biraderi system is a means of attributing social status and providing mutual social support[25]. Some biraderi groups such as Rajput and Jatt have been present on the Indian sub-continent for the past 2000–3000 years[26–28]. Others originated in the early Medieval period, such as the Gujjars whose identity is traced back to the Gurjara kingdom in present-day Rajasthan around 570 C.E.[29]. Mass conversion to Islam during the pre-Mughal and Mughal era introduced multiple new biraderi groups, including Syed, Qureshi, Malik and Sheikh[30]. Historical records suggest that endogamous practices started during the Gupta Empire, strengthened during the Mughal Empire[31], and became even stricter during the colonial times of the nineteenth century, as the social classification system was reinforced by the British to solidify their political authority, enable rationalised taxation, and establish rules about property[31,32]. Overall, however, there are limited historical records about when the biraderi groups emerged or the extent to which endogamy was practiced over the centuries. Very little is known about the effect of this historical social structure on present-day genetics, since previous work has been limited to small studies of a few microsatellite markers[28,33,34].

Here, we analyse genotype array data and exome-sequence data from thousands of Pakistani-ancestry individuals sampled in Britain as part of the BiB project. BiB is a birth cohort set up to investigate the social, environmental, and genetic causes of poor health and educational outcomes of children born in Bradford, a city with high levels of socio-economic deprivation in the north of England[16]. Around half of the individuals in this cohort have Pakistani ancestry, coming primarily from Azad Kashmir (Mirpur) and northern Punjab (see map in Supplementary Fig. 1), in proportions similar to the British Pakistanis as a whole[8,35]. BiB has rich self-reported information on the Pakistani mothers' biraderi groups, places of birth, and parental relatedness. To our knowledge, this is the largest sample of Pakistani-ancestry individuals analysed to date to explore population genetic questions.

## Results

**Samples and dataset**. We assembled a large dataset of 7180 individuals with Pakistani ancestry (Supplementary Data 1 and 2) from the BiB project, of which 5669 had been genotyped on the Illumina CoreExome array and 1511 on the Illumina Global Screening Array (GSA); 2484 of these also had exome-sequence data. Identifying related individuals is challenging in a population with high consanguinity and endogamy, so we tried several algorithms (see 'Methods', Supplementary Fig. 2). Conservatively, we used the algorithm that gave the highest estimates of kinship, PropIBD from KING[36], to identify and remove putative relatives (third-degree or closer). Most analyses in this paper are based on genotype data from 2200 unrelated mothers (CoreExome array; 251,853 single-nucleotide polymorphisms (SNPs) with minor allele frequency (MAF) >1% post-quality control). Some analyses additionally used genotype data from 1616 unrelated children (CoreExome array) or 228 unrelated fathers (GSA array), and exome-sequence data from 2484 mothers (see below). Supplementary Data 18 indicates that samples were used in which analyses.

After cleaning questionnaire responses about biraderi membership, we determined that 56 distinct groups had been reported. Most of these are recognised biraderi groups, but some individuals identified themselves with tribal or regional groups (e.g. Pathan or Kashmiri), or clans within a biraderi (e.g. Choudhry) (Supplementary Data 3). Presuming that individuals have reported the labels that best represent their group identity within the context of the Bradford Pakistani community, we henceforth refer to these collectively as self-reported 'subgroups' rather than 'biraderi'.

**Population structure**. We began by investigating the genetic relationships between the Bradford Pakistanis and other world-wide populations using publicly available datasets, considering autosomal, Y chromosome and mitochondrial markers (Supplementary Note 1, Supplementary Figs. 3, 4 and 19 and Supplementary Data 4, 5, 6, 7 and 15). Comparisons with both modern and ancient genomes suggested that the different Bradford subgroups all come from a common ancestral population with a large ANI component (Supplementary Note 1).

We then examined the fine-scale population structure within the Bradford Pakistanis. Principal component analysis (PCA) of the samples revealed clear structure, with the first three PCs each explaining ~4% of the variation and driven, respectively, by the separation of the Jatt and Choudhry subgroups, the Pathan, and the Bains and a subset of Rajput individuals (Fig. 1). The fact that the Choudhry (dark orange colour) and Jatt (teal colour)

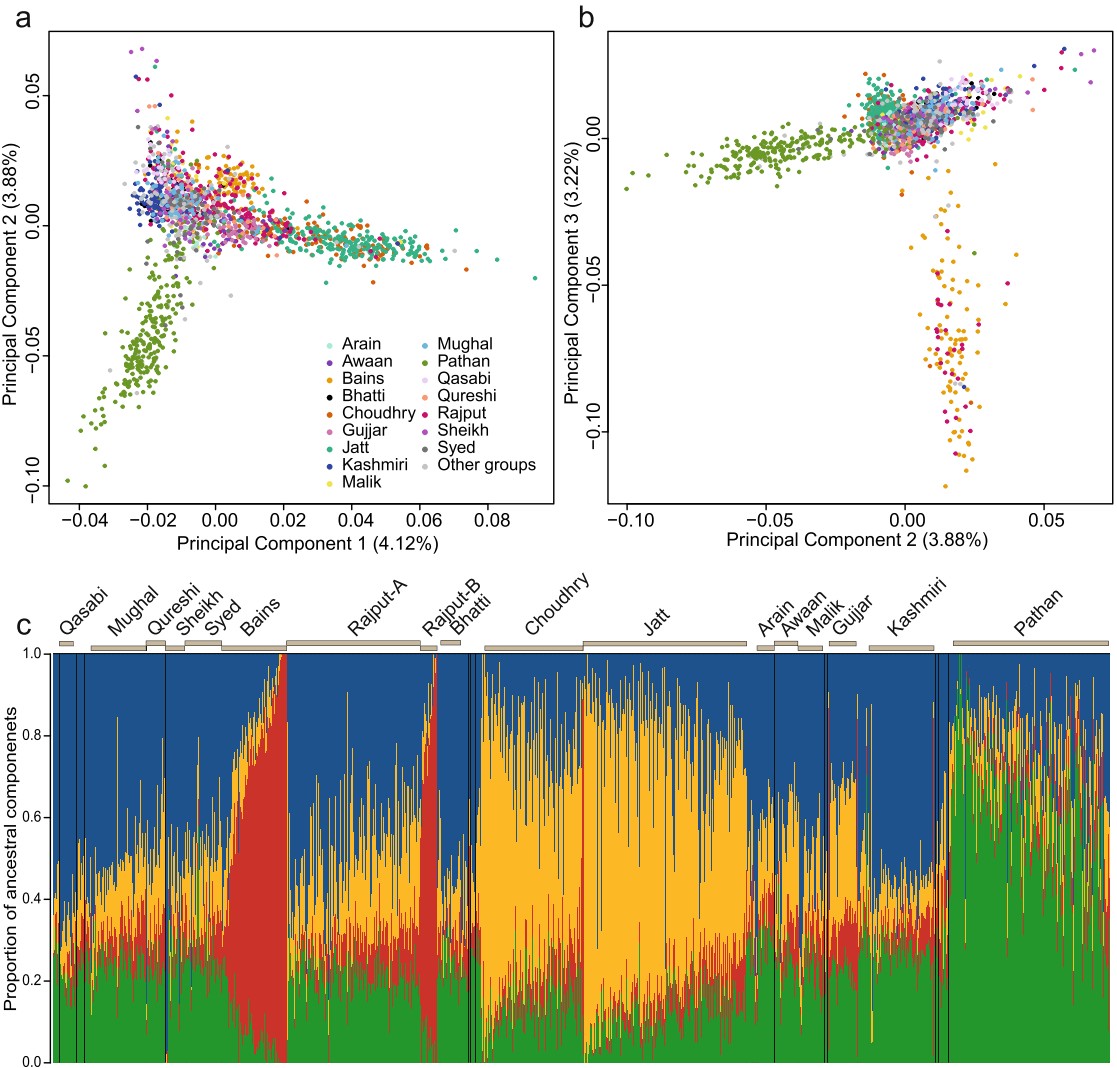

**Fig. 1 Population structure within British Pakistanis from Bradford. a**, **b** Principal components analysis of 2200 unrelated Pakistani mothers, with the self-reported subgroups with >20 samples indicated in different colours. Plots show PC1 versus PC2 (**a**) and PC2 versus PC3 (**b**). Proportion of overall variation explained by each PC is noted in brackets on the axis label. **c** ADMIXTURE plot ($K = 4$) illustrating different ancestral components making up the various subgroups, with the largest self-reported subgroups indicated. We have indicated the two distinct subgroups amongst the Rajput.

subgroups cluster together is consistent with the fact that 'Choudhry' is an honorary title in Punjab and Kashmir used most commonly by the Jatts, who are one of the largest ethnic groups in Pakistan and north-west India[28]. ADMIXTURE analysis of the Bradford Pakistani samples indicates that the subgroups that were distinguishable on the PCA have different proportions of genetic components (Fig. 1c). The Rajput group contains two distinct subgroups that we term henceforth Rajput-A and Rajput-B; the Rajput-B group, which has a higher fraction (>40%) of the 'red' ancestry component, appears more similar to Bains than to Rajput-A, and these individuals cluster with Bains on the PCA (Fig. 1b). We found no evidence that the Rajput-A versus Rajput-B distinction is driven by geographical origin within Pakistan. It may well be that Rajput-B and the different subgroups of Rajput-A represent different sub-clans of Rajputs, of which there are hundreds across South Asia[37,38]. An alternative explanation is that some people self-identifying as Rajputs actually have diverse ancestries since individuals from other subgroups chose to identify with this group to benefit from land allocations during British colonial times[25,39] or to increase their social status after migration to the UK[40]. The observation that

Bains and the Rajput-B group appear to form a homogeneous genetic cluster is consistent with historical evidence that Bains is one of the Rajput families[41].

We next applied Uniform Manifold Approximation and Projection (UMAP) to the first 20 PCs (Supplementary Fig. 5). This allowed us to define the Pathan and the Bains/Rajput-B subgroups more cleanly than on the PCA, but failed to distinguish additional groups. We found no significant correlation (two-sided Mantel test $p$ value: 0.94) between genetic distance (measured by UMAP1 and UMAP2 vectors) and geographic distance (measured by geographic coordinates of the individual's or her parents' self-reported village of origin in Pakistan).

We then applied fineSTRUCTURE[42], a Bayesian clustering algorithm, to a matrix of haplotype-sharing patterns. The inferred hierarchical clustering tree based on 1520 individuals from the 16 major subgroups (Fig. 2) identified three clusters containing the majority of the self-reported Pathan (Cluster 8), Bains and Rajput-B (Cluster 9), and Jatt and Choudhry (Cluster 10) individuals. Subsets of Bains and Rajput-B (Cluster 6) and Jatt and Choudhry (Cluster 11) clustered with individuals from other

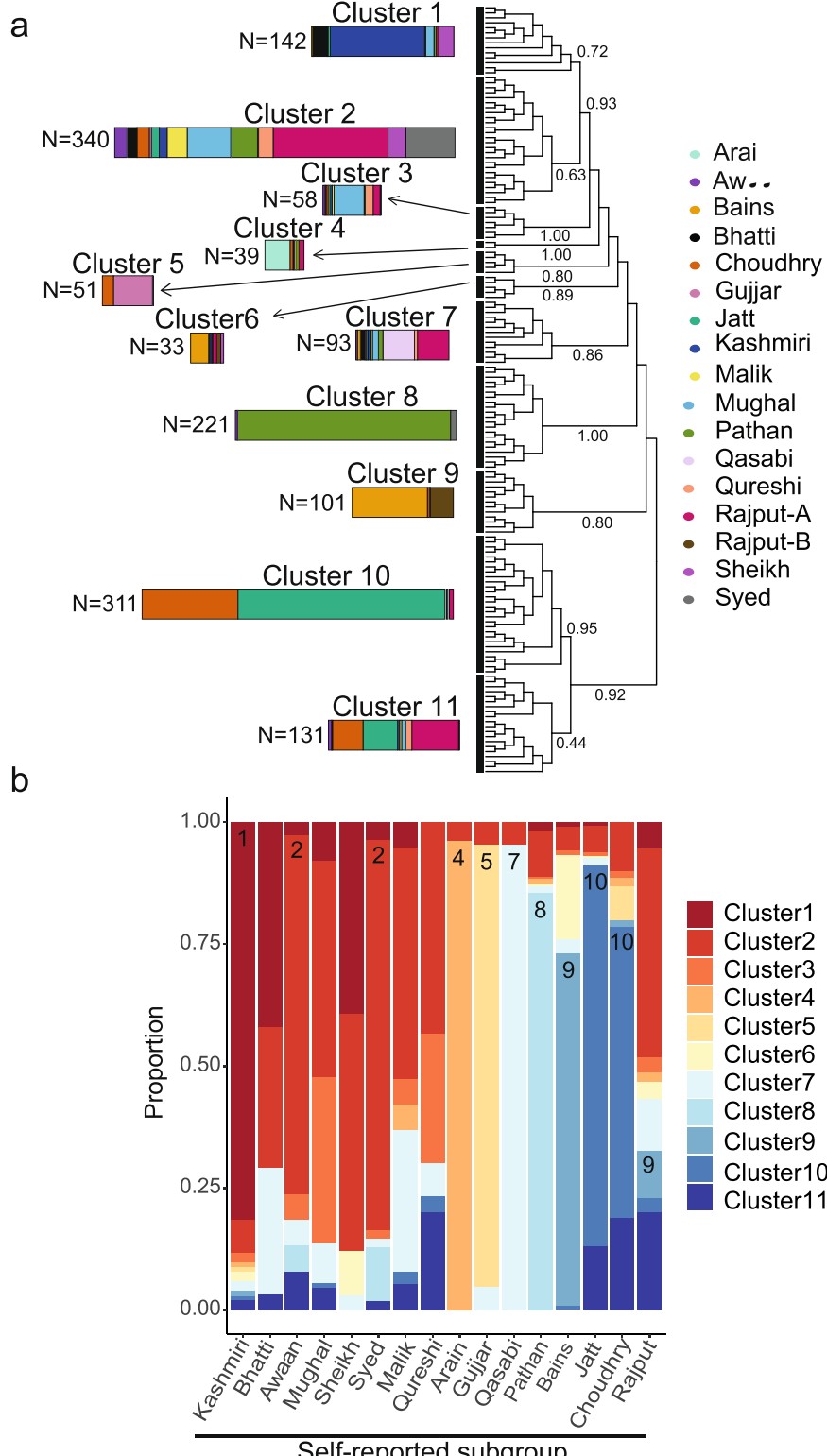

**Fig. 2 Fine-scale population structure inferred amongst 1520 Bradford Pakistani mothers using fineSTRUCTURE. a** The tree illustrates the results of hierarchical clustering of the co-ancestry matrix using patterns of haplotype sharing from ChromoPainter. Each 'leaf' on the tree contains multiple individuals. The coloured bars represent the composition of each of the major clusters indicated by the thick black vertical lines. The length of each coloured bar is proportional to the number of individuals in that cluster, with the proportion of each colour representing the fraction of individuals from each self-reported subgroup that make up the cluster. The labels on the edges are the posterior assignment probabilities from fineSTRUCTURE. **b** The barplot illustrates the fraction of individuals from each self-reported subgroup that fall into the indicated clusters defined by fineSTRUCTURE. The numbers on the barplot indicate the clusters that were used to define a homogeneous subset of individuals from that self-reported subgroup for subsequent demographic analyses.

groups, although Cluster 11 had lower support than the other clusters. The Bains and Rajput-B in Cluster 6 showed a smaller proportion of the red component in the ADMIXTURE plot (Fig. 1c) than those in Cluster 9 (two-sided Wilcoxon's rank-sum test $p$ value $= 1 \times 10^{-12}$). For other self-reported groups, the majority of individuals from the group fell in a single fineSTRUCTURE cluster: the Kashmiri (Cluster 1), Syed and Awaan (Cluster 2), Arain (Cluster 4), Gujjar (Cluster 5), and Qasabi (Cluster 7). Some of the self-reported groups appear to be quite heterogeneous, with individuals from these groups distributed across different inferred clusters; Rajput-A is a notable example. Genetic differentiation was very low ($F_{ST} < 0.001$) between the Awaan and Syed in Cluster 2, the Bains and Rajput-B in Cluster 9, and the Jatt and Choudhry in Cluster 10 (Supplementary Fig. 6 and Supplementary Data 8). We henceforth pooled together individuals in these very similar subgroups who fell within those dominant clusters (Awaan/Syed, Bains/Rajput-B and Jatt/Choudhry), and restricted the Arain, Gujjar, Kashmiri, Pathan and Qasabi subgroups to those individuals who fell within the dominant fineSTRUCTURE cluster for that group. These homogeneous subgroups were recapitulated when downsampling each cluster in Fig. 2a to 60% of its original size and rerunning the fineSTRUCTURE clustering (Supplementary Note 1 and Supplementary Fig. 20), confirming that cryptic relatedness between samples was unlikely to explain the structure we observed.

The biraderi system is culturally characterised by patrilineal kinship ties, so we examined GSA data on 228 unrelated Pakistani fathers from BiB to test whether males from the same subgroups did indeed carry the same or similar Y haplogroups (Supplementary Data 5), using 1056 Y chromosomal SNPs. Although 79% of the individuals carried the IJ haplogroup, and the sample size is relatively small, we did not find a clear delineation of some groups observed in the analysis of the autosomal data. Even males within the most distinct groups (e.g. Jatt/Choudhry, Bains/Rajput-B, Pathan) did not tend to cluster together (Supplementary Fig. 7). This is consistent with previous findings in Punjabi Rajputs[34] and in Jatts[28], and may suggest at least three possible explanations that are not mutually exclusive: the founders of each biraderi may have included males with several different haplogroups, the age of the biraderi groups may not be old enough to lead to strong Y-haplotype stratification, and/or historically the patrilineality of the biraderi system may not have been very strict.

**Demographic history**. We next estimated population divergence times between the homogeneous subgroups defined in the previous section, using the NeON package[43] that considers patterns of linkage disequilibrium (LD) decay. We found that all groups diverged from one another within the last ~70 generations (Fig. 3a, Supplementary Data 9 and Supplementary Data 10), consistent with the low genetic differentiation between the subgroups (Supplementary Fig. 6). When we removed individuals with high autozygosity, the divergence time estimates tended to be shifted towards more recent times, but the relative patterns remained similar across groups (Supplementary Fig. 8 and Supplementary Data 10). The estimates in Fig. 3a suggest that the history of these subgroups cannot be considered as a series of clean splits between ancestral populations; rather, it appears that several groups began to differentiate from one another around the same time, with some degree of admixture persisting between the groups after their initial divergence. It is important to note that admixture is not explicitly modelled when estimating split times, thus depending on the extent of the gene flow in each subgroup, estimates might be more or less biased. Nonetheless, we can see

that Bains/Rajput-B (Cluster 9 in Fig. 2) show the greater divergence time from the other groups, followed by Qasabi (Cluster 7) and Pathan (Cluster 8) (Supplementary Data 9). Greater divergence may be interpreted as either older divergence time between groups or greater genetic drift caused by a small population size. However, simulations under different demographic scenarios showed that NeON infers divergence times robustly in the last 200 generations even in the presence of decreasing $N_e$ (Supplementary Note 1 and Supplementary Fig. 21). Within clusters, Bains and Rajput-B (Cluster 9) have a divergence time estimate close to zero, as do Jatt and Choudhry in (Cluster 10; Supplementary Data 11), consistent with their clustering together on fineSTRUCTURE and having very low $F_{ST}$ ($F_{ST} < 0.001$).

Population relationships and gene flow inferred by Treemix[44] suggested that the Bains/Rajput-B group experience the strongest genetic drift, followed by Qasabi and Pathan (Supplementary Fig. 9a–c). The tree topology without migration edges explained most of the variance (97.4%), but adding one or two migration edges increased the variance explained to >99%, detecting gene flow events from Pathan into Kashmiri (one migration edge), and from Bains/Rajput-B and Jatt/Choudhry into Kashmiri (two migration edges) (jackknife analysis $p$ values <0.001). $f$3-statistics[45] also showed that the Kashmiri underwent significant admixture events with other subgroups (Supplementary Fig. 9d and Supplementary Data 12). Although about half of the Bradford Pakistanis come from Azad Kashmir, people who refer to themselves as 'Kashmiri' are believed to originate specifically from the Kashmir Valley[46]. Our finding that the Kashmiri have gene flow from other groups is consistent with historical evidence that the Kashmiri from the valley belonged to different biraderi groups and tribes. Some migrated to Azad Kashmir and Punjab during the nineteenth century, where they began identifying themselves as Kashmiri ahead of their biraderi identity, tending to marry within this group[41,47,48]. This likely explains the relative genetic homogeneity of the Kashmiri in our sample (80% of individuals fall in Cluster 1 on Fig. 2) and the observation that the most recent estimated divergence time between Kashmiri and other groups is ~10 generations ago (Fig. 3a).

We next applied IBDNe[49] to infer recent effective population sizes ($N_e$) through time across the different subgroups. All subgroups, and indeed, all Pakistanis combined (Supplementary Fig. 10d), were inferred to have relatively low $N_e$ over the last 50 generations compared to white British individuals (Fig. 3b), likely reflecting the endogamy of the biraderi system. Bains/Rajput-B (Cluster 9), Jatt/Choudhry (Cluster 10) and Pathan (Cluster 8) showed a similar trend: a strong reduction in $N_e$ around 10–20 generations ago followed by a recovery. The other subgroups, except for Arain, showed a progressive decrease in $N_e$ in the last 15 generations, which may reflect the increase in endogamy that occurred during the Mughal Empire and British colonial times[31]. Broad conclusions were unchanged in various sensitivity analyses (Supplementary Note 1 and Supplementary Figs. 10 and 11). It is important to note that the high $N_e$ estimates in recent generations might be due to phasing errors and should be thus interpreted with caution. Furthermore, gene flow between the subgroups could also have an impact on the inferred $N_e$ trajectory.

**Endogamy and consanguinity**. To quantify the extent of this historical endogamy, we calculated identity-by-descent (IBD) scores[50] in the Pakistani subgroups. These scores represent the average total length of IBD shared between any two individuals from the same subgroup after excluding relatives, considering segments between 5 and 30 cM. Most of the Pakistani subgroups have substantially higher IBD scores than other worldwide

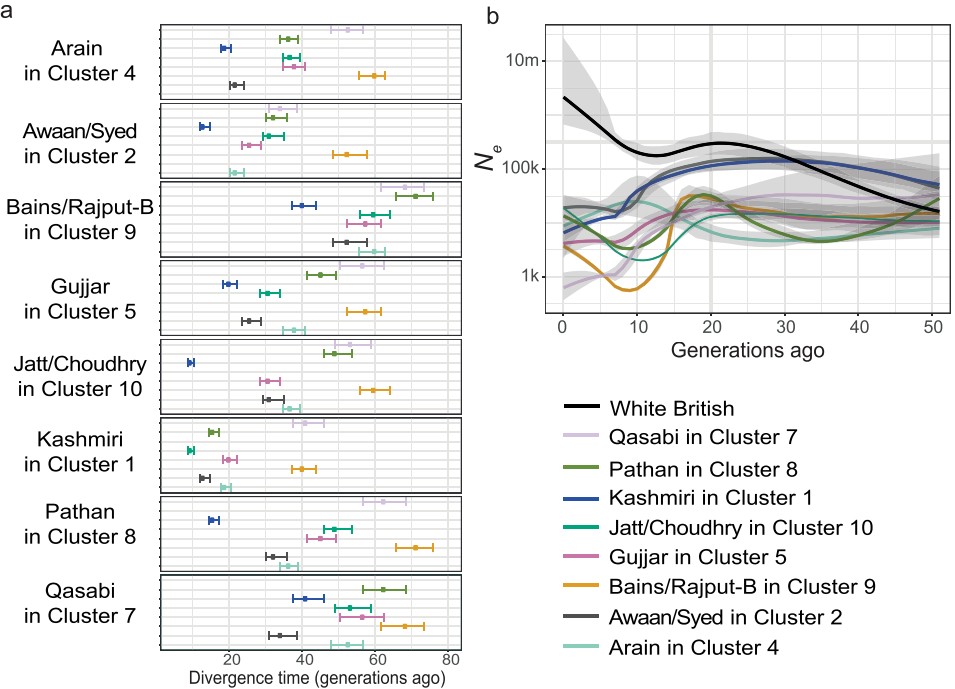

**Fig. 3 Divergence times and historical effective population size changes of Bradford Pakistani subgroups.** Note that these are the homogeneous subgroups defined using the fineSTRUCTURE clusters. **a** Divergence times estimated using NeON[43] ($n = 850$ independent individuals). Within each panel, the points show the estimated divergence time between the group indicated on the left on the y-axis and those indicated in the legend. The horizontal lines indicate 95% confidence intervals of the estimated divergence time. **b** Change in effective population size ($N_e$) through time estimated with IBDNe. The coloured lines indicate the mean estimate and the grey shading indicates 95% confidence intervals of the $N_e$ estimates.

populations from the 1000 Genomes Project, including the Finns (Fig. 4a), and similar to those reported previously for some isolated Indian groups[50]. The Bains/Rajput-B group had the highest IBD score, followed by Qasabi and Jatt/Choudhry. Similar results were obtained in various sensitivity analyses (Supplementary Note 1 and Supplementary Fig. 12).

Fifty-seven per cent of the BiB Pakistani mothers reported that their parents were related, and 63% reported being related to their child's father (Supplementary Data 13 and 14). As expected, a much higher fraction of the genome was homozygous ($F_{ROH}$) in the Pakistani mothers than the White British (mean = 0.048 versus 0.0004, two-sided $t$ test $p < 1 \times 10^{-15}$). Amongst the BiB children, those whose parents were both born in the UK had significantly lower $F_{ROH}$ than those whose parents were both born in Pakistan, and both groups had significantly lower $F_{ROH}$ than those who had one parent born in Pakistan and the other in the UK (Supplementary Fig. 13a). This fits with some prior evidence that cousin unions may be preferred for trans-national marriages[51] among first- and second-generation British Pakistanis, for whom they can be a means of ensuring cultural continuity and socio-economic support, and of allowing siblings separated by migration to reconnect through the marriages of their children

We developed an algorithm to infer parental relatedness based on the frequency and length of ROHs in an individual (see 'Methods' and Supplementary Note 1 for details), similar to previously published approaches[52,53]. Briefly, we simulated different types of consanguineous couples to derive empirical distributions of the frequency of ROHs of different lengths as well as the length of the ten longest ROHs per individual. We then used a neural network trained on the simulated data to predict parental relatedness in our cohort. The model was 92.0% accurate when considering the three classes of the parental relationship shown in Fig. 4b–d. Our results suggest that many families have

been practising cousin marriage for multiple generations (Supplementary Fig. 14a, b). When comparing the self-reported parental relatedness to the inferred parental relatedness, we find that 81% of BiB Pakistani mothers who reported having parents who are first cousins are inferred to have parents who are first cousins or closer (Fig. 4b). However, the self-reporting seemed to be less reliable for the individuals who said their parents were first cousins once removed or second cousins ($z$ test for population proportions: $p = 0.0048$ for first cousins versus first cousins once removed, $p < 1 \times 10^{-15}$ for first cousins versus second cousins) (Fig. 4b). Overall, consanguinity was under-reported in the mothers and slightly over-reported in the children though less so than in mothers, with 78% of the mothers inferred to have parents who are second cousins or closer compared to the 57% reported, versus 58% inferred and 63% reported for the children ($z$ test for population proportions: $p < 1 \times 10^{-15}$ for mothers, $p = 0.0008$ for children, $p < 1 \times 10^{-15}$ for the difference in reporting between mothers and children) (Supplementary Fig. 14).

Rates of consanguinity markedly differed between subgroups (Fig. 4c). Most notably the Qasabi subgroup had 67% of individuals with parents who are unrelated (more distant than second cousins) while nearly all Bains/Rajput-B individuals had parents who were inferred to be second cousins or closer. Even when considering individuals inferred to be offspring of unrelated parents, $F_{ROH}$ differed significantly between the subgroups (Fig. 4d). Notably, groups that had higher $N_e$ in recent generations including Arain, Awaan/Syed and Kashmiri (Fig. 3b) had overall lower $F_{ROH}$, suggesting weaker endogamy in these groups. To examine the contribution of endogamy and consanguinity to variation in $F_{ROH}$ within our cohort, we fit a joint model on inferred consanguinity and subgroup defined using fineSTRUCTURE. Inferred consanguinity explained 62% of

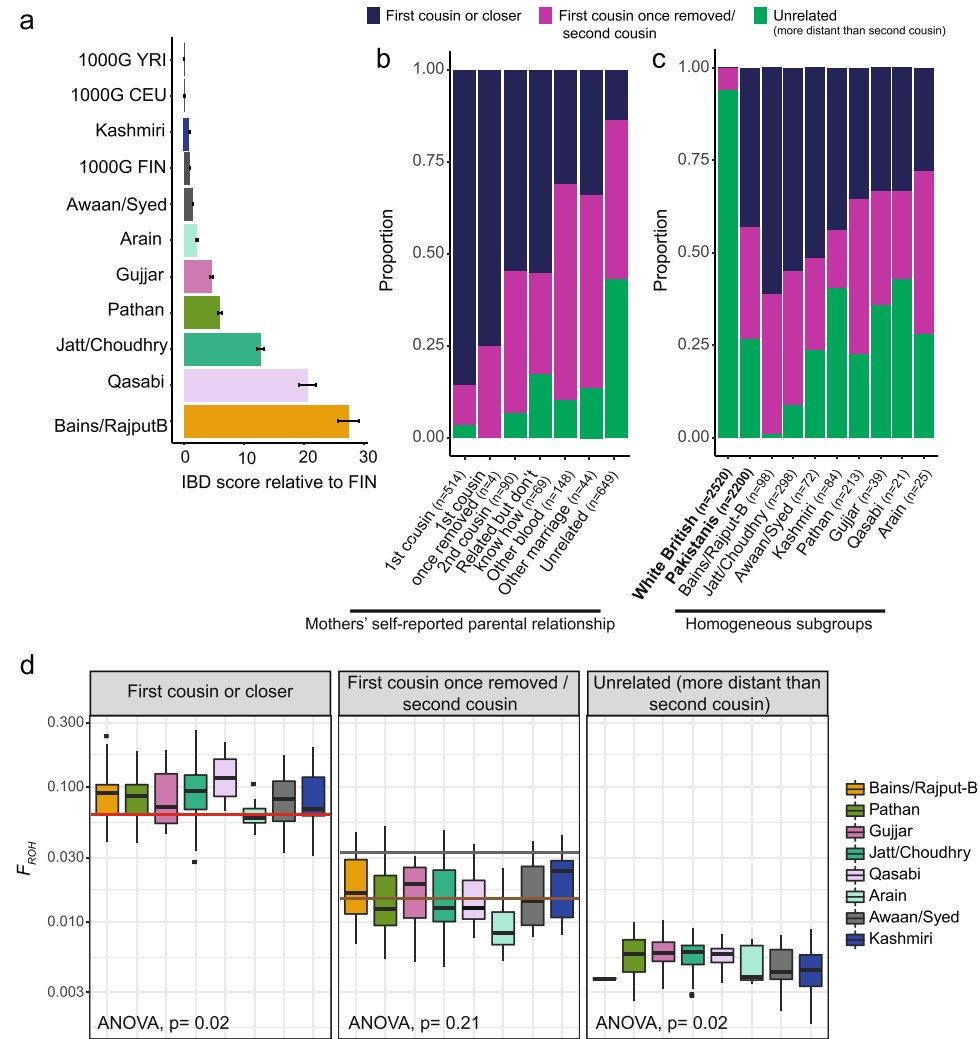

**Fig. 4 Effect of endogamy and consanguinity on IBD sharing and ROH patterns in BiB Pakistani subgroups.** Note that these are the homogeneous subgroups defined using the fineSTRUCTURE clusters. **a** IBD scores were calculated as the average total length of IBD segments between 5 and 30 cM shared between individuals from the indicated group ($n = 964$ independent samples), standardised by the value for the 1000 Genomes Finns (FIN). The error bars indicate the IBD score standard error. YRI Yoruba, CEU Western Europeans. **b, c** Stacked bar plots showing inferred parental relatedness **b** in the Pakistani mothers broken down by self-reported parental relatedness or **c** in all white British versus all Pakistani mothers, and in Pakistani mothers broken down by subgroup. **d** Boxplots showing the fraction of the genome in regions of homozygosity ($F_{ROH}$) broken down by subgroup and inferred degree of parental relatedness ($n = 850$ independent samples). In each boxplot the centre is equal to the median, the upper and lower bounds of the box correspond to 25th and 75th percentiles and the whiskers represent 1.5 times the inter-quartile range (IQR) from the bounds of the box. Outliers are represented as points. Lines indicated expected $F_{ROH}$ for individuals with parents who are first cousins (red), first cousins once removed (grey) and second cousins (brown). The two-way ANOVA tests whether $F_{ROH}$ differs between the subgroups.

the variance in $F_{ROH}$ (two-way ANOVA $p < 1 \times 10^{-15}$), subgroup explained 5% (two-way ANOVA $p < 1 \times 10^{-15}$), and the interaction term was not significant (two-way ANOVA $p = 0.17$). The pattern of differences in $F_{ROH}$ across subgroups was similar between individuals who were born in Pakistan and those born in the UK (Supplementary Fig. 13b), implying that each subgroup's relative frequency of consanguineous unions has not changed substantially after migration.

To validate our inference of the degree of parental relatedness for the Bradford Pakistanis using an orthogonal method, we applied recently developed theory[54], which predicts the length distribution of ROH segments given historical consanguinity rates and a historical $N_e$ trajectory. We focused on mothers from the two largest homogeneous subgroups: the Pathan from Cluster 8 ($N = 213$) and the Jatt/Choudhry from Cluster 10 ($N = 299$)

(Fig. 2). Our neural net predicted an average kinship between the mothers' parents of 0.035 for Pathan and 0.057 for Jatt/Choudhry (see 'Methods'), versus values of 0.016 and 0.028 estimated based on self-reported parental relationships. Our neural net estimator did not explicitly take into account the possibility that some ROH segments may descend from a more remote common ancestor even in the absence of any close relationship between the parents. However, it did focus only on ROHs >10 cM, which Supplementary Fig. 15 illustrates are minimally impacted by $N_e$, except when it is very small ($N_e < 5k$). To verify that the $N_e$ trajectory was not affecting our inferences from the neural net, we examined the expected 'ROH footprint'[54] given the $N_e$ estimates from IBDNe (Fig. 3b) (see 'Methods'), and given consanguinity rates derived either from self-reported information or based on our neural net. Figure 5 shows that the observed ROH (and IBD) footprints are

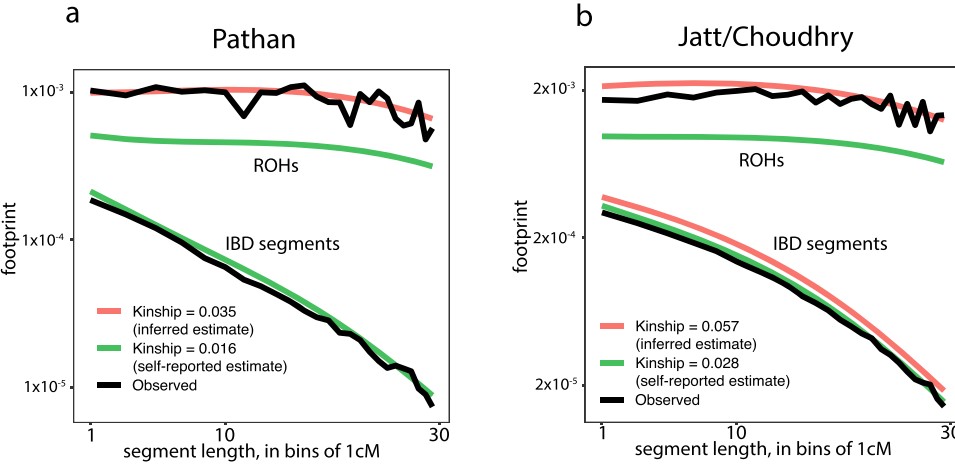

**Fig. 5 Observed ROH and IBD footprints compared to the expectation under a coalescent model[54].** The footprint is the average fraction of the genome covered by segments of a given length interval. The top lines represent the ROH footprint and the bottom lines the IBD footprint. Points are plotted at the beginning of each 1 cM interval. The $N_e$ profile used to compute the expected footprints was determined by IBDNe using **a** the Pathan from fineSTRUCTURE's Cluster 8 or **b** Jatt/Choudhry from Cluster 10. In **a**, for the IBD segments, the pink line is beneath the green one. We used kinship values calculated using either the self-reported relationships or the inferred estimates of consanguinity for the relevant group. The observed footprints (black lines) were determined using filtered IBD and ROH calls from IBDseq and bcftools/roh, respectively. Supplementary Fig. 16 shows equivalent plots using the upper and lower bounds of the 95% confidence interval for the IBDNe estimate as the $N_e$ trajectory, and using observed footprints from different ROH or IBD calling/filtering strategies.

similar to those expected when using the kinship values inferred from our neural net-based method rather than the self-reported information, confirming the robustness of our inference. Similar results were obtained with ROHs called by PLINK, whereas GARLIC seems to call more short ROHs than the model predicts (Supplementary Fig. 16), consistent with the higher false-positive rate for short ROHs reported in the original GARLIC paper[55].

Our results suggest that, even in the absence of close consanguinity, increased homozygosity due to endogamy is likely to be contributing to recessive disease burden[56] and the elevated frequency of rare homozygous knockouts[14,57] in this population. To investigate the relative impact of endogamy versus consanguinity on recessive disease risk, we used exome-sequence data from 2484 Bradford Pakistani mothers, in which we ascertained pathogenic/likely pathogenic (P/LP) variants in autosomal recessive developmental disorder genes. We then simulated intra-biraderi (endogamous) and inter-biraderi (exogamous) couples, and unions between pairs of individuals whose IBD distribution matches that of self-reported first cousins within the dataset (see 'Methods'). We then scored each couple as being 'at risk' of having an affected child if both individuals were carriers of a P/LP variant in the same gene, similar to the approach in ref. [58]. The results (Fig. 6) indicate that intra-biraderi unions incur significantly higher risk than inter-biraderi unions (particularly for the Bains and Jatts; one-sided permutation tests $p = 2 \times 10^{-4}$ and $p < 1 \times 10^{-4}$ respectively), but first cousin unions incur more than ten-fold higher risk than intra-biraderi unions.

## Discussion

We have carried out the first large-scale investigation of the population structure and demographic history of Pakistanis. We found genetic structure in the cohort reflecting the influences of the biraderi social stratification system, with some subgroups forming identifiable and homogeneous clusters (Figs. 1 and 2). Our analyses suggest that these subgroups come from a common ancestral population, but diverged from one another within the last 70 generations (1500–2000 years) (Fig. 3a). This is consistent

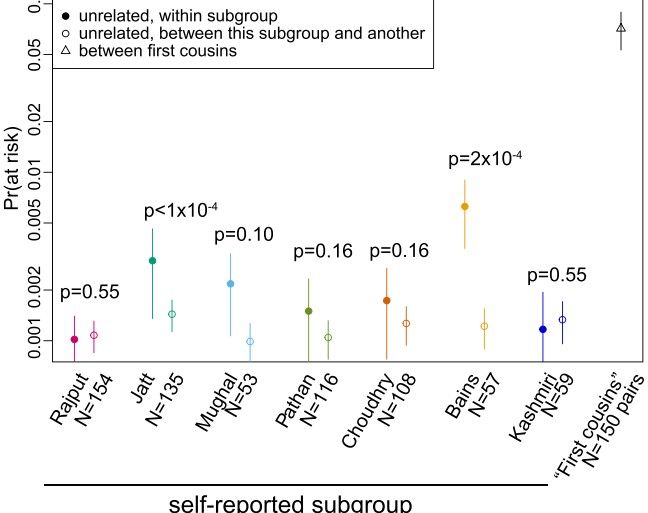

**Fig. 6 Estimated fraction of couples from intra-biraderi, inter-biraderi and first cousin unions who would be at risk of having a child with a recessive developmental disorder.** These estimates are based on simulations that use P/LP variants ascertained from exome-sequence data in Bradford Pakistani mothers, and rely on the self-reported subgroup information. The y-axis indicates an estimate of the fraction of couples that would be 'at risk' of having a child with a recessive developmental disorder since both parents are carriers of P/LP variants in the same gene. Note that this is undoubtedly an under-estimate of absolute risk (see 'Discussion'), but one can still compare the relative risk for intra- versus inter-biraderi unions. Error bars indicate standard deviations of the risk estimates and one-sided p-values are from permutation tests (see 'Methods'). The 'first cousins' were sampled to match the IBD distribution observed for self-reported first cousin couples (see 'Methods'), regardless of biraderi group.

with an earlier finding that the transition from intermarriage to strict endogamy on the Indian subcontinent started from about 70 generations ago[20], concurrent with or immediately after the drafting of the ancient Law Code of Manu that described a ranked social stratification system[32].

Historically, intra-biraderi marriages and consanguinity were practised to solidify socio-economic bonds[59,60]. Intra-biraderi marriages continue to be very common in the Bradford Pakistani community, constituting >90% of marriages in the Gujjar, Pathan, Jatt, and Bains, ≥80% in the Syed, Qasabi, Rajput, Awaan, and Choudhry, and ~60–63% in Qureshi, Malik, and Sheikh in the current BiB participants[60], according to the self-reported questionnaire data. As one might expect, the subgroups we inferred to be the most genetically homogeneous and with the highest IBD sharing (Fig. 3a) are generally those with the highest self-reported rates of intra-biraderi marriage. Endogamous practices became stronger, particularly amongst the elite classes, during the Mughal Empire (mid-1500s to mid-1800s)[31], and then amongst all classes under the British Raj (mid-1800s to 1947) when the laws of land ownership changed[31,32]. This could be an explanation for the decrease in effective population size seen in all subgroups starting 15–20 generations ago (~375–580 years ago if the average generation time were 25–29 years) (Fig. 3b). However, there is considerable uncertainty in the average generation time; according to historical records, it was not uncommon at some periods in history and in some regions of the world for women to marry and have children as early as 12–18 years old, including in South Asia[61,62]. Hence, we should be cautious about attributing these changes in $N_e$ to demographic changes or historical events at particular time points.

To assess the rates of consanguinity in the cohort, we developed a neural net-based algorithm to infer the degree of parental relatedness based on ROH patterns in individuals. We found that rates of consanguinity were under-reported in the mothers and varied significantly between biraderi groups, highlighting differential social practice within the British Pakistani population. Consanguinity rates seemed to be reported somewhat more accurately for the children (Supplementary Fig. 14), possibly because the mothers were more sure of their genetic relationship with their husbands than of that between their parents. We found evidence of multiple generations of recent consanguinity in many families (Supplementary Fig. 14), which contribute to many individuals having higher $F_{ROH}$ than expected given the self-reported parental relationships. Our data also suggest that endogamy has contributed to differences in $F_{ROH}$ between biraderi groups (Fig. 4d). These results could potentially inform prior expectations about recessive disease risk for British Pakistanis in a clinical genetics setting[56]. It should be noted that there could be misclassifications with our neural net-based method due to systematic bias in the arbitrary set of modelled consanguineous relationships. Additionally, there is significant variability in the ROH distribution for a given consanguineous relationship, further complicating accurate classification due to overlapping ROH distributions.

To validate our findings from this neural net-based method, we leveraged new coalescent-based theory[54] (and 'Methods') to predict ROH levels due to recent consanguinity, while, for the first time, properly accounting for ROH attributed to historical endogamy. We found that for both the Jatt/Choudhry and Pathan subgroups, a naive estimate of spousal kinship based on self-reported parental relatedness underestimates the proportion of the genome in ROH segments, whereas an estimate of kinship based on the inference from our neural net gives an expectation very similar the observed data. This approach can in principle also detect historical changes in the consanguinity rates; however, we found little power to distinguish different levels of historical consanguinity rates (Supplementary Fig. 17).

We showed through simulations based on exome-sequence data that intra-birarderi unions increase the risk of having offspring with rare recessive disorders, but less so than first cousin unions (Fig. 6). The elevated risk was particularly pronounced in the groups that showed the strongest degree of bottleneck in the IBD score analysis. We emphasise that this analysis assumes full penetrance, and that the variants included are the only ones that cause recessive disease in this population, and hence underestimates absolute risk, but this seems less of a problem if we consider the estimates of relative risk. However, we note that our analysis ignored missense and inframe variants that are pathogenic but have not been reported with two stars in ClinVar. Such variants are likely to be rarer and more recent and thus contribute more to risk for intra-biraderi than inter-biraderi marriages. For this reason, our estimates of the relative risk may be considered lower bounds. Future work should consider larger sample sizes, more exhaustively identify all potentially pathogenic recessive variants and evaluate the risks of endogamy versus consanguinity in cohorts of rare disease patients rather than in these simulations.

Our findings suggest that clinicians should consider recording parents' biraderi groups as well as close relatedness in genetic consultations. This will be particularly useful as research becomes more focused on clinical sequencing datasets such as that held by Genomics England. Recording biraderi information would enable further research into the prevalence of different diseases in different biraderi groups, the impacts of endogamy and the possible presence of disease-causing founder mutations. The results from such research will be important to inform and design targeted genomic health services for Pakistani-ancestry populations. However, great care needs to be taken to ensure this research and any application of it is carried out in a culturally sensitive way. (See Broader Impact section below.)

This study has several limitations. Although we had samples from 56 distinct groups, many of them had a small sample size that would not allow reliable group-level inference about demographic history or founder events. The majority of our sample comprised individuals with Pathan, Punjabi or Kashmiri ancestry residing in the UK; additional data from the UK and Pakistan, including from other ethnic groups such as the Baloch and Sindhi, would allow us to explore how generalisable our findings are. Furthermore, our study was based primarily on SNP-genotype data, which, although highly valuable for describing population structure, did not allow the finer-scale demographic inference that would be possible with a large sample of whole-genome sequence data. For example, whole-genome sequence data on a larger sample size of males would allow us to leverage more Y chromosomal markers to further explore whether biraderi membership has indeed been passed down patrilineally in recent times. Finally, it is challenging to accurately estimate relatedness in populations with high consanguinity and endogamy. Multiple lines of evidence suggest that our approach has correctly removed individuals who are third-degree relatives or closer (Supplementary Figs. 2c, e and 4e) and that our identification of homogeneous clusters was not affected by the accidental inclusion of relatives (Supplementary Fig. 20). However, we cannot exclude the possibility that we have been over-cautious in removing putative relatives. This may mean we have over-estimated recent $N_e$ and under-estimated $F_{ST}$ between groups, leading to underestimation of the divergence time. We also note that the admixture between the groups has likely affected our estimates of historical $N_e$ and divergence time.

We have presented the largest investigation to date of fine-scale population structure and history in Pakistanis. We have shown how the biraderi social stratification system has played a significant role in shaping the population structure in Pakistani communities over the last 70 generations. Endogamous practices have led to greatly elevated IBD sharing as well as increased homozygosity, which is likely to have implications for disease risk on top of the high rates of consanguinity. Larger sample sizes and

data on disease diagnoses (as opposed to proxy estimates of risk) will be needed to quantify the risk of recessive disorders for offspring of intra- versus inter-biraderi unions accurately, noting that our results imply that this differs between biraderi groups, and that certain disorders may be enriched in particular biraderi groups as a result of founder effects. Furthermore, future studies of disease genetics in Pakistanis should consider our findings in order to best control for stratification due to recent population structure in genome-wide association studies[3] and potentially to increase power by exploiting the IBD within some groups as a proxy for rare variant sharing[63].

**Broader impact**. There is an urgent need to expand genetic research to populations with non-European ancestry so that more people will benefit in the future from precision medicine. An understanding of population structure is an important pre-requisite for robust medical genetic studies and can be used to inform study design to ensure validity and maximise power. This study explores the genetic structure of a population with Pakistani ancestry, which has a high burden of particular diseases, but has been, until recently, largely neglected in genetic research. Our findings can be used to inform future medical genetic studies of Pakistani-ancestry individuals in the UK and around the world. For example, our work motivates further exploration of how best to control for recent, fine-scale population structure when conducting genome-wide association studies in this population and of whether polygenic risk scores perform differently in different Pakistani sub-populations.

Our work touches on some sensitive topics, such as how people's genetic ancestry may differ from their cultural identity, and how the degree of parental relatedness inferred from people's genetic data may differ from their own beliefs or understandings. Our findings of the elevated risks of rare recessive disorders associated with intra-biraderi (endogamous) marriage have the potential to be mis-interpreted and lead to stigmatisation if not appropriately communicated. Nevertheless, we hope that this risk is outweighed by the benefits that may be reaped from future medical genetics research in the Pakistani community that builds on our findings. At each stage of this research, we have consulted with representatives from the Bradford Pakistani community and, with their input, we have prepared a set of key questions and answers to accompany the paper (Supplementary Note 2), to try to present our results and their implications clearly and sensitively to a lay audience.

Further research would be required in population-based cohorts containing rare disease patients to calculate absolute risks arising from consanguineous and intra-biraderi marriages. Estimating absolute risks would be more clinically useful than the relative risks we have presented here. We note that previous work from the BiB study showed that advanced maternal age confers a similar level of increased risk of congenital anomalies to that associated with consanguinity[12]. Our findings motivate a search for recessive disease-causing founder variants in particular biraderi groups, the discovery of which would enable carrier and/or prenatal testing for variants causing severe disorders. A culturally competent approach and sensitive communication strategy will be required to carry out such research and to translate these results into health education and clinical practice, in order to maximise patient benefits and to ensure all groups are well informed and empowered to make informed decisions regarding their marriage choices[64]. A joint clinical and social science approach alongside a community engagement pro-gramme would help to build trust within the British Pakistani population, improve access to genomic medicine services and provide a means to reduce health disparities. Similar approaches to other preventive health interventions like obesity[65] and smoking cessation[66] were acceptable for delivery among British Pakistanis.

## Methods

**Dataset**. The data come from the Born in Bradford (BiB) study. We complied with all relevant ethical regulations for work with human participants. Ethical approval for the data collection was granted by Bradford Research Ethics Committee (ref. 07/H1302/112), and informed consent was obtained from participants. All women (of any ethnicity) giving birth at the Bradford Royal Infirmary in 2007–2010 were invited to participate in the study and ~80% of those eligible accepted[16]. We had genetic data from 7180 individuals of Pakistani ancestry and 6818 of White British ancestry (Supplementary Data 1 and 2) from the BiB cohort. These were pre-dominantly from mothers and their children, with some fathers also included. The samples were genotyped using two chips: (1) the Infinium CoreExome-24 v.1.1 BeadChip (~550 K SNPs), and (2) the Infinium GSA-24 v.1 (~640 K SNPs). We also analysed whole-exome sequencing (WES) data for 2484 Pakistani individuals who were also genotyped. Recruited mothers were asked to fill in a self-reported questionnaire, including, for the Pakistani mothers, questions about biraderi groups, places of birth, and parental relatedness (consanguinity).

**Quality control of genotype chip data**. Data quality control was performed using PLINK v.1.90b4[67]. We required <1% missingness per SNP and <10% missingness per individual. Duplicated variants were removed. We removed 110 samples with sex discrepancies, 123 genetic duplicates and 465 individuals who were not genetically related to someone who should have been a first-degree relative, and five individuals who had an inferred first-degree relative who should not have been related. Genetic ancestry was assigned with PCA using the self-reported infor-mation on ethnicity, and an additional 52 samples were filtered out because their declared ethnicity was different from their genetic ancestry inferred using EIGENSOFT 7.2.1[68,69]. SNPs with Hardy–Weinberg equilibrium $p$ value $<1 \times 10^{-6}$ were removed considering Pakistani (3348 CoreExome variants, 1435 GSA var-iants) and White British (305 CoreExome variants, 224 GSA variants) separately. These filters resulted in a dataset of 476,816 autosomal SNPs (246 mitochondrial DNA (mtDNA) SNPs) and 14,624 individuals on the Core Exome chip and 598,326 SNPs (583,667 autosomal, 1056 Y chromosome) and 4398 samples on the GSA. For this study, we focused primarily on individuals of Pakistani ancestry, but in some figures, the White British individuals were used for comparison. Results presented are from the CoreExome chip data, unless otherwise stated, since this gave a larger sample size, and taking the intersection of the two chips would have left insufficient SNPs for most analyses.

We performed the demographic inferences on mothers genotyped on the CoreExome chip using common autosomal SNPs (251,853 SNPs with MAF > 0.01). For the PCA and ADMIXTURE analyses, we filtered out SNPs in high LD ($r^2 > 0.5$). Fathers from the GSA dataset were used for the Y-chromosome analysis. This resulted in a dataset of 3081 Pakistani and 2873 white British mothers and 2588 Pakistani children on the CoreExome chip and 601 mothers and 235 Pakistani fathers on the GSA.

**Inference and removal of relatives**. A critical step in population genetic analysis is identifying and removing close relatives. In endogamous populations, distin-guishing recent familial relatedness from population structure is challenging because both show genetic similarity through allele sharing[70]. We inferred relat-edness coefficients with KING 2.2.4, which has been reported to be robust to population structure[36], unlike PLINK's $\hat{\Pi}$ estimator. We compared the original kinship estimate from KING (called 'KING-robust' in ref. [36]) to a new estimator in KING, PropIBD, which integrates IBD segment information to infer sample rela-tionships for third- and fourth-degree relatives more accurately (http://people.virginia.edu/wc9c/KING/manual.html). Comparisons of these different estimators on the mothers genotyped on the CoreExome showed that PropIBD identified more third-degree and closer relatives than the other methods (Sup-plementary Fig. 2a, b): KING ProbIBD removed 881 samples, KING-kinship removed 741 samples and PLINK's $\hat{\Pi}$ removed 821 samples.

To try to determine which of these estimators was more accurate, we compared the genetic estimates of kinship to self-reported relationships for 196 Pakistani mother–father pairs for whom the relationship had been declared by the mother on the questionnaire. For these, since mothers and fathers had been genotyped on different arrays, we took the intersection of CoreExome chip and GSA (134,218 SNPs with MAF > 0.01, with SNP missingness <1% in the merged set) and ran KING on this (Supplementary Fig. 2c, d). The results suggest that KING-kinship may be underestimating true kinship for third-degree relatives: 21/75 (28%) self-declared first cousins were called more distant than third-degree relatives by KING-kinship, versus only 4/75 (5%) with KING-PropIBD. PropIBD called 38/75 (51%) self-declared first cousins as second-degree relatives. This seems to be mostly driven by inbreeding in the previous generation, since when we restrict to couples for whom both the mother's and father's parents were reported to be unrelated, KING-kinship and KING-PropIBD gave very concordant estimates (Supplementary Fig. 2d).

To be conservative in our population genetic analyses, we decided to use the relatedness estimates from KING-PropIBD to exclude relatives. We removed one sample from each pair of individuals with third-degree relatedness or above (i.e. PropIBD>0.0884). We retained 2200 Pakistani and 2520 White British mothers and 1616 Pakistani children on the Core Exome chip and 544 Pakistani mothers and 228 Pakistani fathers on the GSA.

As further validation of KING-PropIBD and to assess whether the estimates were robust to the endogamy in the cohort, we simulated individuals of various degrees of relatedness using the real-phased genotype data from individuals who self-declared as coming from the same biraderi and who fell in the same fineSTRUCTURE cluster (Fig. 2a). Specifically, we took the data phased with Eagle v.2.4.1 and used the recombination map provided with Eagle to generate gametes and combined them along a given pedigree using custom R code. We generated 500 pairs of individuals with each of four relationships: siblings, first cousins, second cousins, and unrelated. Reassuringly, KING-PropIBD inferred the correct or a closer relationship in the vast majority of cases; all sibling pairs and 93% of first cousin pairs were inferred to have the true simulated relationship or closer (Supplementary Fig. 2e).

**Biraderi categorisation**. On the questionnaire, Pakistani mothers in BiB were asked to state their own biraderi, that of each of their parents and that of their husband and of his parents. We cleaned the biraderi membership data, checking for spelling errors and combining groups with variable spellings of the same group. We determined that 56 distinct groups had been reported. To ensure we were assigning individuals to the biraderi that was most consistent with their parental biraderi, we compared the mother's self-reported biraderi to the biraderi reported for each of her parents, and likewise for the father's biraderi. The biraderi for a mother or father was set to missing if it was not reported ($N = 645$), or if her/his parents' biraderi was not reported ($N = 58$) or was discordant with his/her own ($N = 118$). The biraderi for a child was assigned following the same approach used for mothers and fathers. We assigned 2324 mothers (1652 unrelated), 1568 children (906 unrelated) and 169 fathers (164 unrelated) unambiguously to biraderi groups.

**Population structure**. For comparison with other modern worldwide populations, we merged our dataset with the HGDP[71] and 1000 Genomes Project Phase 3[22]. For comparison with modern and ancient individuals, we combined our samples with a dataset of published genotypes from modern and ancient individuals (https://reich.hms.harvard.edu/datasets)[18]. In both cases, we used GRCh37.

PCA was performed on the pruned datasets using EIGENSOFT 7.2.1[68,69]. For the worldwide dataset, the eigenvectors were computed using the non-BiB datasets and the individuals from BiB were projected onto them (Supplementary Fig. 3a, b). For Supplementary Fig. 3c, to ensure that the BiB Pakistanis were not dominating the structure due to their large sample size, we computed the PCA using HGDP Pakistani populations and just 25 BiB Pakistanis, then projected the remaining BiB Pakistanis onto them.

ADMIXTURE v.1.3[72] was run on the pruned datasets and the cross-validation error was calculated for identifying the best $K$ value, which was found to be 4 for Fig. 1c and 8 for Supplementary Fig. 4a. We separated Rajput-A and Rajput-B according to the proportion of the red component in the ADMIXTURE analysis, defining Rajput-B individuals as those for which the red component made up >40% of their ancestry. Rajput-A individuals had a maximum red component proportion of 18%.

Genetic affinity with other ancient and modern worldwide populations was tested by computing $f$-statistics using ADMIXTOOLS v.6.0. We computed both $f3$- and $f4$-statistics. Outgroup $f3$-statistics were computed with the phylogeny $f3$(Bradford subgroups, ancient samples; Mbuti), where the ancient samples were representative individuals of different archaeological periods and locations in South and Central Asia[18]. We selected ancient samples with the highest number of SNPs overlapping with our cohort. We also computed outgroup $f3$-statistics using present-day populations with the phylogeny $f3$(Bradford Pakistanis, X; Mbuti), where X represents another worldwide population (Supplementary Data 3). We considered both all Bradford Pakistanis together and the self-reported subgroups separately. We computed $f3$-statistics with the phylogeny $f3$(X,Y; Bradford subgroup), where X and Y are other Bradford subgroups (Supplementary Fig. 9d). Standard errors were obtained using blocks of 500 SNPs. $f4$-statistics were computed using qpDstat with f4mode:YES and with the phylogeny $f4$(W, X; Y, Chimpanzee), where W represents the biraderi groups reporting Arabic ancestry (Qureshi, Sheikh or Syed), X are all the other Bradford subgroups and Y represents Middle East populations[45] (Supplementary Data 5). Positive values indicate gene flow between W and Y.

We used qpgraph in the ADMIXTOOLS v.6.0 package to confirm that the Bradford subgroups could be modelled as a mixture of ANI and ASI ancestral components as seen in ref. [18]. For this analysis, we considered the homogeneous subgroups defined using fineSTRUCTURE (Fig. 2a) to avoid biases in the proportion estimates. We used the same qpgraph parameters and populations used in ref. [18]. We assessed the fit of the demographic model for each of the Bradford subgroups separately, and estimated the proportion of the ANI and ASI components in each subgroup. Differences in the empirical and theoretical $f$-statistics with Z-score ≥3 were considered statistically significant.

UMAP was computed from the top 20 PCs with default parameters in the BiB dataset using the *uwot* R package[73]. To compare genetic to geographic distance, we first assigned 547 mothers to a village of origin in Pakistan using either the mother's own self-reported village of origin or that of her parents if she was born in the UK and reported that her parents were born in the same Pakistani village. We then determined the latitude and longitude of the villages in order to define geographic distances between them. Genetic (measured as UMAP1 and UMAP2 vectors) and geographic distances were computed as Euclidean distances using the dist R function with method = Euclidean. Correlation between genetic and geographic distance was estimated using a two-sided Mantel test implemented in the *Ade4* R package (mantel.rtest function)[74]. The number of permutations used for the Mantel test was 9999.

We ran fineSTRUCTURE v.4.0.1[42] to infer fine-scale population structure in the BiB Pakistani mothers. The haplotypes were phased using SHAPEIT v.2.12[75] using the 1000 Genomes Phase 3 dataset as a reference panel. We generated the co-ancestry matrix using ChromoPainter and used it to run fineSTRUCTURE with 1,000,000 burn-in steps and 1,000,000 iterations. A PCA on the co-ancestry matrix using the *prcomp* R function confirmed the structure in the tree (Fig. 2). We ran the algorithm twice on the dataset with the major subgroups to make sure the tree structure was robust (data not shown). We also ran it on the dataset including all subgroups to confirm that the clusters were consistent.

Given the heterogeneity of the genetic clusters inferred in our cohort (Fig. 2), we defined self-reported groups as homogeneous if at least 60% of the individuals from each self-reported subgroup fell in the same genetic cluster inferred by fineSTRUCTURE. We then estimated pairwise $F_{ST}$ (calculated with the Weir and Cockerham's method using the program 4P[76,77]) between homogeneous self-reported groups within a fineSTRUCTURE cluster, and combined two groups if they had $F_{ST} < 0.001$ (5th percentile of the empirical distribution for all pairs of groups shown in Supplementary Fig. 6b). For demographic analyses, we included only the homogeneous self-reported groups that had at least 20 individuals. These were: Awaan+Syed from Cluster 2, Bains+Rajput-B from Cluster 9, Jatt +Choudhry from Cluster 10, Arain from Cluster 4, Gujjar from Cluster 5, Kashmiri from Cluster 1, Pathan from Cluster 8 and Qasabi from Cluster 7. Sample sizes for these groups are shown in Supplementary Data 8.

Chromosome Y haplogroups were defined with yhaplo[78] and mtDNA haplogroups with Haplogrep2[79]. A median-joining haplotype network for the Y-chromosome data from BiB Pakistani fathers was constructed with PopART v.1.7[80].

**Divergence time estimation**. We calculated the time of divergence between the homogeneous sub-populations and genetic clusters using the approach described in McEvoy et al.[81] and implemented in the NeON R package[43]. The 95% confidence intervals for the times of divergence were calculated using a jackknife procedure, leaving out one chromosome each time.

**IBD segment calling**. We conducted several analyses using IBD segments between the Pakistani individuals, called using IBDseq v.r1206[82] and/or GERMLINE 1.5.3[83]. IBDseq requires unphased data, whereas GERMLINE phased haplotypes. IBDseq ran with default parameters. For GERMLINE, we phased the data with Eagle v.2.4.1[84] using 1000 Genomes Project Phase 3 genetic maps. IBD segments were defined using GERMLINE with the parameters -bits 75 -err_hom 0 -err_het 0 -min_m 3 -h_extend, and the HaploScore algorithm[85] was used to remove false-positive IBD tracts (genotype error = 0.0075, switch error = 0.003, mean overlap = 0.8). We did some additional filtering of IBD calls as described in Supplementary Note 1.

**Estimating historical effective population size**. We estimated historical changes in effective population size ($N_e$) for both Pakistani subgroups and White British individuals using IBDNe v.19Sep19.268[49]. As recommended, we ran IBDseq[82] to identify IBD segments with default parameters. We then excluded IBD chunks overlapping problematic regions as explained above, although in practice we found this had a minimal effect on the results. We ran IBDNe with the default parameters, except that we set a lower limit of 5 cM for the IBD segments considered. We also ran IBDNe using the IBD calls from GERMLINE and found that this gave systematically higher $N_e$ estimates than IBDseq, although the trajectories were very similar, particularly for the larger groups (Supplementary Note 1 and Supplementary Fig. 10). We ran IBDNe with different sets of samples and filtering to confirm the robustness of our estimates (Supplementary Note 1 and Supplementary Figs. 10 and 11).

**Calculating IBD scores**. We computed the IBD score developed in Nakatsuka et al.[50] from GERMLINE IBD calls to quantify the extent of founder events in each population using IBD segments. In addition to the aforementioned removal of individuals who were third-degree relatives or closer based on the KING-PropIBD metric (PropIBD > 0.0884), we also removed one individual from each pair of samples sharing an IBD chunk >40 cM. We calculated IBD scores as the total length of IBD segments between 5 and 30 cM detected between individuals in the same subgroup divided by the total number of possible pairs, i.e. $[\binom{2n}{2}) - n]$, where $n$ is the number of samples in that subgroup. We then standardised each IBD score

by the score for the Finnish individuals from 1000 Genomes Phase 3[50]. Standard errors for IBD scores were calculated using a weighted block jackknife for each chromosome, and 95% confidence intervals were defined as the IBD score $\pm 1.96 \times$ standard error. We also computed IBD scores using the same set of filters used in Nakatsuka et al. (IBD segment filter >30 cM to identify relatives, then counting IBD segments between 3 and 20 cM; Supplementary Fig. 18) and found that the results were similar (Supplementary Fig. 12a). We found similar results with different sensitivity analyses (Supplementary Note 1 and Supplementary Fig. 12). We also computed the IBD scores using IBD calls from IBDseq using the same strategy used for GERMLINE IBD calls (Supplementary Note 1 and Supplementary Fig. 12e).

**Inferring gene flow between Pakistani subgroups.** Gene flow was inferred using the Treemix v.1.13 approach[44] and $f3$-statistics[19,45]. We ran the Treemix analysis with default parameters, without the -root option. We added migration edges (-m) until we reached a total variance explained of 99.5%. The $f3$-statistics were calculated in the form of $f3$(target, source 1, source 2) using ADMIXTOOLS v6.0, and provide evidence that the target population is derived from an admixture of populations related to sources 1 and 2. We tested all possible combinations of target and source populations in our dataset. Standard errors were obtained using blocks of 500 SNPs. Tests with a $Z$-score $< -3$ were considered significant.

**ROH calling.** We used three different ROH callers: bcftools/roh 1.13, GARLIC v1.6.0a, and PLINK. The results in Figs. 4 and 5 and Supplementary Figs. 13, 14, 15 and 17 are based on bcftools/roh calls, and in Supplementary Fig. 16 we compare all three callers.

With bcftools/roh, we used the -G 30 flag. We noticed an excess of apparently artefactual ROHs spanning long gaps between SNPs around centromeres, so we removed ROHs <10 Mb overlapping centromeres. In practice, this made very little difference to the ROH footprint (Supplementary Fig. 16). For GARLIC[55,86], we set the number of resamples for estimating allele frequencies (--resamples) to 20 and we assumed a genotyping error rate (--error) of 0.001. We used the --auto-winsize flag to guess the best window size based on the SNP density. For PLINK, we followed previous publications[87,88] and used the following parameters: --homozyg-window-snp 50 --homozyg-snp 50 --homozyg-kb 1500 --homozyg-gap 1000 --homozyg-density 50 --homozyg-window-missing 5 --homozyg-window-het 1.

**Inferring degree of parental relatedness from ROH patterns.** In order to infer the degree of parental relatedness of individuals for whom parental genotype information was unavailable, we simulated pedigrees with various types of consanguineous unions and used a neural network classifier to infer parental relatedness from the patterns of homozygosity in the offspring, similar to approaches used in ref. [52,53]. We used the data phased with Eagle v2.4.1. Using the subset of individuals inferred to be unrelated from the original dataset, we used the recombination map provided with Eagle to generate gametes and combined them along a given pedigree that involved a particular type of consanguineous union. We considered ten different classes of consanguineous union: sibling, avuncular unions lasting for up to three generations, first cousin unions lasting up to three generations, first cousin once removed, second cousin and unrelated.

For each type of consanguineous parental relationship, we simulated 500 pedigrees using custom R code to obtain empirical distributions for the length of ROHs over 10 cM in individuals whose parents had that relationship. We called ROHs on simulated and real individuals using bcftools/roh as described above. We used 15 statistics for the purposes of classification using the neural net: the total length of the ten longest ROHs (in cM), and the frequency of ROHs ranging from 10 to 150 cM binned into 14 intervals of 10 cM. Using these statistics, we trained a neural net classifier implemented in the R package *nnet* to assign individuals to a given consanguineous pedigree. When testing the algorithm, we trained the model on 400 individuals from each class and predicted the assignment of 100 individuals from each class of the ten different parental relationships. We tested the algorithm by randomly sampling individuals ten times, each time simulating an equal number of individuals from each of the ten model classes. Each of the ten times we assigned an individual to a parental relatedness category based on the category for which they got the most classifications across the ten trials. The accuracy was determined as the fraction of individuals classified correctly by the neural net.

We conducted a $z$ test for population proportions to compare the proportion of individuals inferred to have parents with a particular relationship with the proportion who self-reported that relationship. Similarly, we used a $z$ test for population proportions to assess whether the difference in the fraction who were inferred to be versus self-reported as consanguineous differed between mothers and children.

Further details regarding the accuracy and robustness of the method are available in Supplementary Note 1. Code is available at https://github.com/malawsky/consanguinity_simulation.

**Analysis of $F_{ROH}$.** $F_{ROH}$ was determined for each individual by summing the lengths of the (autosomal) ROHs in base pairs and dividing by the length of the autosomal genome. The figures given in the 'Results' section come from bcftools/roh calls, but very similar results were obtained with PLINK and GARLIC (data not shown). We calculated $F_{ROH}$ in the BiB mothers and children from the bcftools/roh calls and stratified these estimates according to the parents' birthplace (both parents born in the UK, one parent born in the UK and one in Pakistan, and both parents born in Pakistan), by the degree of parental relatedness inferred as described in the previous section and by subgroup.

**Analysis of ROH and IBD distributions.** We applied theory from refs. [54,89] to determine the expected length distribution and genomic footprint of ROHs and IBD segments given a particular historical $N_e$ trajectory and rate of consanguinity. In the model developed in the above papers, there are $N_e$ pairs of parents, who, in each generation, have probability $r_1$ of being (full) siblings, $r_2$ of being (full) first cousins, $r_3$ of being second cousins, etc., up to a certain degree $n$. A key parameter of the model is the average kinship between parents,

$$k = \sum_{i=1}^{n} \frac{r_i}{4^i}, \tag{1}$$

which is the probability of a random chromosome in each parent coalescing due to consanguinity. It is assumed that the parents are related only through a single path of degree 1, ..., $n$, or are otherwise unrelated.

In ref. [54], a Markov chain was derived for the evolution of the state of two chromosomes. The possible states are as follows: the chromosomes are in unrelated individuals, in one of each parent, in the same individual as two homologous chromosomes, or in the same individual as a single chromosome (coalescence). We then considered $t_{between}$, the time to the most recent common ancestor (TMRCA, or coalescence time) of two chromosomes sampled from two unrelated individuals, and $t_{within}$, the TMRCA of the two chromosomes of an individual. In refs. [54,89], the mean and variance of $t_{between}$ and $t_{within}$ were computed using a first-step analysis. An approximation of the distribution was derived using a separation-of-time-scales analysis, The results showed that $t_{between}$ is approximately exponential with rate $1/[4N_e(1-3k)]$. The factor of 4 is because there are $N_e$ pairs of parents, thus $2N_e$ individuals, and $4N_e$ chromosomes. $t_{within}$ has probability $k/(1-3k)$ of being $O(1)$, i.e. representing a rapid coalescence within the family due to consanguineous unions, and is otherwise approximately exponential as $t_{between}$.

Here, we assume that $t_{between}$ is distributed as in the standard coalescent, but with a population size trajectory scaled according to the theory as $N_e(t)(1-3k)$. Thus, in discrete time, it has the distribution

$$P(t_{between} = t) = \frac{1}{4N_e(t)(1-3k)} \prod_{\tau=1}^{t-1} \left(1 - \frac{1}{4N_e(\tau)(1-3k)}\right). \tag{2}$$

For $t_{within}$, as the approximate distribution groups together all rapid coalescence events, we used a composite approach. For $t \leq 50$, the distribution was computed numerically by running the exact Markov chain for 100,000 iterations. For $t > 50$, the distribution was set to

$$P(t_{within} = t > 50) \propto \frac{1}{4N_e(t)(1-3k)} \prod_{\tau=1}^{t-1} \left(1 - \frac{1}{4N_e(\tau)(1-3k)}\right) \tag{3}$$

(as for $t_{between}$), where the coefficient of proportion was set such that the entire distribution was normalised. Running the exact model for only $t < 10$ or $t < 40$ gave almost indistinguishable results.

We defined the genomic 'footprint' of IBD and ROH segments as the proportion of the genome (in genetic map units) found in segments (of each type, respectively) of length between $[l_1, l_2][90]$. Given the distributions $P(t_{between})$ and $P(t_{within})$, we computed the footprint as follows. In ref. [91], Ringbauer et al. (Eq. (4) therein) showed that, in a chromosome of length $L$, the mean number of segments with TMRCA $t$ of length in the small interval $[l, l+\Delta l]$ (in Morgan) is:

$$E[n_{seg}(l; L, t)] = 4te^{-2tl}(1 + t(L-l)) \cdot \Delta l \tag{4}$$

The mean chromosome length covered by segments (with TMRCA $t$) of length in $[l_1, l_2]$, denoted $c(l_1, l_2; L, t)$, is:

$$E[c(l_1, l_2; L, t)] = \int_{l_1}^{l_2} E[n_{seg}(l; t)] l \, dl = e^{-2tl_1}(L + 2Ltl_1 - 2tl_1^2) - e^{-2tl_2}(L + 2Ltl_2 - 2tl_2^2). \tag{5}$$

The mean length covered by segments of all TMRCAs is obtained by numerically summing over all TMRCAs, using their distributions ($t_{between}$ for IBD segments, who are between unrelated individuals, and $t_{within}$ for ROH segments):

$$E[c_{IBD}(l_1, l_2; L)] = \sum_{t=1}^{\infty} E[c(l_1, l_2; L, t)] P(t_{between} = t), \tag{6}$$

$$E[c_{ROH}(l_1, l_2; L)] = \sum_{t=1}^{\infty} E[c(l_1, l_2; L, t)] P(t_{within} = t). \tag{7}$$

Finally, the footprint is obtained by summing over all chromosomes and dividing by the total genome length,

$$E[f(l_1, l_2)] = \sum_{i=1}^{22} E[c(l_1, l_2; L_i)] / \left(\sum_{i=1}^{22} L_i\right), \tag{8}$$

where $L_i$ is the length of chromosome $i$, in Morgans. The code implementing this model can be accessed at https://github.com/scarmi/ibd_roh.

We set the values for $r = (r_1, r_2, r_3, r_4, r_5)$ in two different ways:

1. Using the self-reported parental relatedness: we set $r_1 = 0$ and $r_5 = 0$ and estimated $r_2$ and $r_3$ using the fraction of mothers who declared that their parents were first cousins/first cousins once removed or second cousins, respectively. We set $r_4$ to be the proportion of mothers who declared that their parents were related but that the relationship type was unknown or was described as 'other blood' or 'other marriage'. This gave an estimate for $r = (r_1, r_2, r_3, r_4, r_5)$ of (0, 0.235, 0.052, 0.155, 0) for Pathan in cluster 8 and of (0, 0.425, 0.057, 0.151, 0) for Jatt/Choudhry in cluster 10, giving naive kinship estimates 0.016 and 0.028, respectively.

2. Using the parental relatedness inferred with our neural net method: we calculated the parameters using the number of individuals inferred to be offspring of the following relationships, as follows:

$$r_1 = \frac{\# \text{ sibling}}{\# \text{ individuals}} \quad (9)$$

$$r_2 = \frac{\# \text{ first cousin (1st generation)} + \# \text{ first cousin (2nd generation)} + \# \text{ first cousin (3rd generation)}}{\# \text{ individuals}} \quad (10)$$

$$r_3 = \frac{\# \text{ first cousins once removed} + 2 \times \# \text{ first cousins (2nd generation)} + 2 \times \# \text{ first cousins (3rd generation)} + \# \text{ second cousins}}{\# \text{ individuals}} \quad (11)$$

$$r_4 = \frac{3 \times \# \text{ first cousin (3rd generation)}}{\# \text{ individuals}} \quad (12)$$

This is because, e.g. someone who is the offspring of first cousins and whose parents and grandparents were also offspring of first cousins is also the offspring of second cousins by two routes, and third cousins by three routes. Note that it is thus possible to have the sum of the components of $r$ being >1. The model still holds as long as the kinship satisfies $k < 1/4$. For Jatt, we obtained the following estimates:

$$r_1 = 0.02341137$$

$$r_2 = 0.08361204 + 0.14053512 + 0.28093645 = 0.5050836$$

$$r_3 = 0.13043478 + 2 \times 0.14053512 + 2 \times 0.28093645 + 0.23411371 = 1.207492$$

$$r_4 = 3 \times 0.28093645 = 0.8428094$$

For Pathan, we obtained the following estimates:

$$r_1 = 0.007389671$$

$$r_2 = 0.111455399 + 0.079812207 + 0.126150235 = 0.3174178$$

$$r_3 = 0.111455399 + 2 \times 0.079812207 + 2 \times 0.126150235 + 0.286384977 = 0.8097653$$

$$r_4 = 3 \times 0.126150235 = 0.37845069$$

To investigate the effect of a sudden change in consanguinity rates at a certain point in the past, we tried reducing the kinship value to $1 \times 10^{-4}$ at the time the distribution switched to the approximate model ($t = 50$). However, this proved to have only very subtle effects on the expected ROH and IBD segment footprint (Supplementary Fig. 17), so we could not evaluate this possibility with the observed data.

For the $N_e$ trajectory, we used a constant value of $N_e$ for Supplementary Fig. 15. For the other analyses, we used trajectory estimated with IBDNe for the last 50 generations (using the point estimate for Fig. 5 and the bounds of the 95% confidence interval for Supplementary Fig. 16), followed by a constant value for $t > 50$ (the IBDNe estimate at $t = 50$). [We divided the IBDNe population sizes by 2 before plugging them into the model, as the model assumes $N_e$ is the number of reproducing pairs.] We note that by using IBD data alone, IBDNe estimates $N_e(t)(1 - 3k)$ rather than $N_e(t)$. Thus, given the IBDNe inferred trajectory $\hat{N}_e(t)$, we set

$$4N_e(t)(1 - 3k) = \hat{N}_e(t) \quad (13)$$

in the equations above.

To compute the ROH footprint in the real data, we averaged the total lengths of ROH segments (within each length interval) over all individuals in the group, and divided by the number of individuals. For the IBD footprint, we averaged the total lengths of IBD segments over all pairs of individuals in the group, and divided by $2n(2n - 2)/2$ (where $n$ is the number of individuals), which is the number of chromosome pairs in different individuals.

We restricted the real data analysis to ROHs and IBD segments greater than 5 cM, since we suspected that segments shorter than this could not be called reliably with the CoreExome chip data, and less than 30 cM, since there were few ROHs longer than this so the average footprint became very noisy. We used 1 cM segment length intervals for both ROH and IBD, where each data point was plotted at the beginning of each interval. The empirical IBD footprint was plotted using the

IBDseq calls filtered as described above, since these calls were used as input for IBDNe.

**Quality control of exome-sequence data.** WES data were generated and mapped using BWA-MEM[92] to the GRCh37+decoy reference genome used by the 1000 Genomes project (ftp.1000genomes.ebi.ac.uk/vol1/ftp/technical/reference/phase2_reference_assembly_sequence/hs37d5.fa.gz). Variant calling was carried out with GATK HaplotypeCaller[93] restricting to the Agilent V5 exome bait regions ± a 100-bp window on either end as described in Narasimhan et al.[14]. Two thousand three hundred and eleven samples were generated as part of this previous publication and had a median on-target coverage of 38×, and 473 additional samples were sequenced later to a median coverage of 50×. This gave a dataset of 2784 individuals, of which 2484 were Pakistani. We excluded variants that failed these criteria, based on standard GATK annotations:

- SNPs: QD < 2, FS > 60, MQ < 40, MQRankSum < −12.5, ReadPosRankSum < −8 or Hardy–Weinberg equilibrium $p$ value $<1 \times 10^{-6}$.
- Indels: QD < 2, FS > 200, ReadPosRankSum < −20 or Hardy–Weinberg Equilibrium $p$ value $<1 \times 10^{-6}$.

We set genotypes to missing if they had genotype quality <20, allelic depth binomial $p$ value <0.001 or depth <7, and then removed variants with more than 20% missingness. This resulted in a dataset of 1,931,299 variants (1,895,447 SNPs and 35,852 indels). The variants were annotated with the Variant Effect Predictor (version 95) using the LOFTEE plugin.

**Simulations to assess relative risk incurred by endogamy and consanguinity on recessive disease.** We began by identifying P/LP variants in autosomal recessive developmental disorder genes from the Developmental Disorder Gene-to-Phenotype (DDG2P) list. We focused on protein-truncating variants annotated as 'high confidence' by LOFTEE that fell in a gene with a loss-of-function mechanism according to DDG2P (915 genotypes at 561 variants in 298 genes), and missense/inframe variants that were classed as pathogenic in ClinVar with two stars and above (264 genotypes at 88 variants in 62 genes).

We then created all possible pairs of exome-sequenced individuals in the dataset, and scored them as 'at risk' if both individuals in the pair carried a P/LP variant in the same gene[58,94]. For a given pair of individuals $i$ and $j$, we write $s_{i,j} = 1$ if they are at risk, and $s_{i,j} = 0$ otherwise. We compared the average $s_{i,j}$ (i.e. the fraction of at-risk couples) between couples that involved two unrelated (i.e. more distant than third-degree) individuals from the same biraderi (intra-biraderi) versus two unrelated individuals from different biraderi groups (one from an index biraderi and the other from any other biraderi). For this analysis, we used the self-reported biraderi information to define the groups, rather than the results from the fineSTRUCTURE analysis, since we wanted to mimic the choices that individuals presumably make when selecting a partner, which are based on self-reported identity. We restricted to groups with at least 50 individuals. We calculated the variance of the fraction of at-risk couples as follows. For couples involving two individuals $i$ and $j$ from the same biraderi of size $n$, we calculated:

$$\text{Var}\left(\frac{\sum_{i \neq j} s_{i,j}}{\frac{n(n-1)}{2}}\right) = \frac{\frac{n(n-1)}{2}\text{Var}(s_{i,j}) + n(n-1)(n-2)\text{Cov}(s_{i,j}, s_{i,k})}{\left(\frac{n(n-1)}{2}\right)^2} \quad (14)$$

This equation assumed that $\text{Cov}(s_{i,j}, s_{k,l}) = 0$, i.e. no correlation in the risk status of distinct couples. The terms $\text{Var}(s_{i,j})$ and $\text{Cov}(s_{i,j}, s_{i,k})$ in the above equation were computed empirically from the data. The variance term was computed as the sample variance of all possible values of $s_{i,j}$. The covariance term was computed as the sample covariance between all pairs of terms $s_{i,j}$ and $s_{i,k}$ that share an index (in other words, all pairs of the risk status of one individual with two other potential mates). For couples involving two individuals, one of whom ($i$) comes from a given biraderi of size $n_1$, and the other of whom ($j$) comes from a different biraderi (total sample size $n_2$), we similarly calculated:

$$\text{Var}\left(\frac{\sum_{i=1}^{n_1} \sum_{j=1}^{n_2} s_{i,j}}{n_1 n_2}\right) = \frac{1}{(n_1 n_2)^2}\text{Var}\left(\sum_{i=1}^{n_1} \sum_{j=1}^{n_2} s_{i,j}\right)$$
$$= \frac{n_1 n_2 \text{Var}(s_{i,j}) + n_1 n_2(n_1 + n_2 - 2)\text{Cov}(s_{i,j}, s_{i,k})}{(n_1 n_2)^2} \quad (15)$$

To obtain an empirical $p$ value for the (positive) difference between the average $s_{i,j}$ for intra- versus inter-biraderi couples, we permuted the biraderi labels 10,000 times, re-estimated the mean $s_{i,j}$ for each permutation, and calculated the fraction of permutations for which the difference mean($s_{i,j,\text{intra}}$) − mean($s_{i,j,\text{inter}}$) was greater than or equal to the observed value.

To estimate risk for first cousin unions, we randomly sampled 150 pairs of individuals 1000 times, matching their kinship (KING-PropIBD) distribution to the observed kinship distribution for BiB Pakistani husband–wife pairs who self-declared as being first cousins (as seen in Supplementary Fig. 2c), in eight bins of unit size 0.05. For example, the proportion of self-reported first cousins with PropIBD between 0.1 and 0.15 was 20/75 (26.67%), so we sampled 40 individuals amongst those 150 randomly chosen pairs to have PropIBD in this range. Each

individual was only included in one pair in a given iteration. The variance for the simulated first cousin pairs was calculated over these 1000 sets of bootstrapped couples.

This analysis makes several assumptions. The most important is that the variants included in the analysis are the only ones that could cause autosomal recessive disorders in this population. This is clearly not true, because we are only considering variants for which there is strong evidence that they are likely to be pathogenic (e.g. excluding novel missense variants). This means that we are underestimating the absolute risk of recessive disorders in this population. However, since these biases should affect all biraderi groups equally, it is likely that our estimates of relative risk for intra- versus inter-biraderi unions are valid. Our analysis also assumes that these variants are fully penetrant when present in a homozygous or compound heterozygous state, and that the random pairing of unrelated individuals accurately recapitulates patterns of marital choices in this community.

**Reporting summary**. Further information on research design is available in the Nature Research Reporting Summary linked to this article.

## Data availability
The Born in Bradford genetic data and questionnaire data (covering self-reported consanguinity, village of origin and biraderi group) are available under restricted access in order to comply with NHS Research Ethics requirements. Access can be obtained as described here https://borninbradford.nhs.uk/research/how-to-access-data/.

The publicly available sources used in this work can be accessed at: https://reich.hms.harvard.edu/allen-ancient-dna-resource-aadr-downloadable-genotypes-present-day-and-ancient-dna-data.

## Code availability
The code for inferring parental relatedness is available at https://github.com/malawsky/consanguinity_simulation (https://doi.org/10.5281/zenodo.5554699).

The code implementing the model of IBD and ROH distributions can be accessed at https://github.com/scarmi/ibd_roh (https://doi.org/10.5281/zenodo.5554362).

The code for the IBD scores was provided by Nathan Nakatsuka and David Reich and will be made available upon request to them.

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

## Acknowledgements

The study is only possible because of the enthusiasm and support of the children and parents recruited and the health professionals and researchers who make Born in Bradford happen, so we thank those individuals. Additionally, we thank Shahid Islam from the Bradford Institute for Health Research, two local British Pakistani leaders in Bradford, Mr. Ishtiaq Ahmed and Mr. Nazir Tabassum, and James Caron from the London School of Economics for their insights into the historical and ethnographic background of the biradari groups. We are grateful to Nathan Nakatsuka and David Reich for the IBD scores code, to Alan Bittles and Noah Rosenberg for their comments on the manuscript, to Alissa Severson and Noah Rosenberg for assisting with the ROH analysis and to Saharon Rosset for statistical advice. We thank Vagheesh Narasimhan for his help with the modelling of ANI and ASI components using qpgraph. This work was supported by a Wellcome core grant to the Wellcome Sanger Institute (098051). S.C. was supported by the United States-Israel Binational Science Foundation Grant 2017024.

## Author contributions

E.A., D.S.M., M.M., T.T. and H.C.M. analysed the data. K.A.H., Q.Q.H., D.M., D.A.v.H. and M.M.I. carried out quality control on the data. S.A.D. and N.S. provided input on the biradari ethnography and history, and S.M.S. and E.S. on the potential clinical implications. J.W. provided the data. S.C. contributed code and intellectual input on the IBD and ROH analyses. M.M.I. and H.C.M. supervised the study. E.A. and H.C.M. wrote the manuscript, with particular input from S.C., D.S.M., M.M., S.A.D. and N.S. All authors commented on the final manuscript.

## Competing interests

The authors declare no competing interests.
