## [Peer Review File · Nature Communications]

Title: Fine-scale population structure and demographic history of British PakistanisREVIEWER COMMENTS

Reviewer #1 (Remarks to the Author):

This manuscript describes analyses of British Pakistanis and particularly presents an investigation of fine-scale population structure and history in Pakistanis. The data and results are of interest. However, I do have some major concerns which I hope the authors can address in a revised version.

(1) It is challenging to conduct traditional population genetic studies in a population with high consanguinity and endogamy because random sampling of “un-related” individuals is not guaranteed. Unfortunately, however, most statistical methods applied in this study assume variables to be independent. Analyses at a “fine-scale” are especially affected and many results&conclusions could be invalidated. For instance, clustering analysis, estimation of divergence times and historical effective population size changes, etc.

(2) In admixture analysis, was $K=4$ the best-supported value for K , based on the cross-validation error? If not, this would indicate potential substructure within the populations that could be of interest, as some more clusters were presented later using other analyses.

(3) In the Abstract, the description “Here, we investigate fine-scale population structure, history and consanguinity patterns using genetic data from >4,000 British Pakistanis.” is not informative. The authors should make it very clear that they analyzed “genotype array data and exome sequence data”, and the exact sample size for each type of the data.

(4) It turned out that the maximal sample size of unrelated subjects, assuming the authors' applied correct analysis, was 2,200. For example, according to the main text, “Most analyses in this paper are based on 2,200 unrelated mothers genotyped on the CoreExome array (~251,853 SNPs with minor allele frequency >1% post-quality control), with 1,616 unrelated children (CoreExome) and 228 unrelated fathers (GSA) used in some analyses.” Therefore, the “>4000” in the Abstract is an extremely exaggerating number, the authors should not overstate the effective sample size (2,200) for real analysis, which is otherwise misleading.

(5) It does not make sense that the authors “assembled a large dataset of 7,180 individuals” for the purpose of this study. The Illumina CoreExome array of 2,200 “un-related” mothers are the only useful data for this paper. Therefore, I would suggest the authors use only these 2,200 or effectively 1,520 “un-related” mothers for this study. Moreover, another advantage of this data set would be to possibly reduce the potential batch effects stemmed from a combination of data generated by different technologies or platforms.

(6) Taken together, I think the authors should better claim a sample size of 2,200 or 1,520 for this paper. The following description in the text is misleading. “We assembled a large dataset of 7,180 individuals with Pakistani ancestry (Supplementary Tables 1 and 2) from the BiB project, of which 5,669 had been

genotyped on the Illumina CoreExome array and 1,511 on the Illumina Global Screening Array (GSA); 2,484 of these also had exome-sequence data.” Because some analyses based on a small number of males might be not reliable, or the conclusion might be misleading.

(7) Sample size can be a key factor for comparative analyses. For example, the following results need some further analysis with random sampling individuals of similar sample size. “However, the self-reporting seemed to be less reliable for the individuals who said their parents were first cousins once removed or second cousins (Figure 4b). Overall, consanguinity was under-reported in the mothers and slightly over-reported in the children, with 78% of the mothers inferred to have parents who are second cousins or closer compared to the 57% reported, versus 58% inferred and 63% reported for the children (Supplementary Figure 14).”

(8) As the authors were also aware, one limitation of the study is a small sample size that would not allow reliable group-level inference about the demographic history of founder events. However, uneven/non-random sampling of the individual's under-representing diversity might be more problematic in population genetic studies. The authors should take caution in interpreting their results.

Reviewer #2 (Remarks to the Author):

This manuscript provides a detailed and novel analysis of fine-scale population structure, endogamy and consanguinity among British Pakistanis in the BiB study. Overall, I think the work will interest many readers of Nature Communications, though I think there are some changes I think the authors could make to more directly highlight the relevance of their work. The most important of these is to explicitly discuss in the abstract the results of Figure 6, which show that endogamy (as quantified by intra vs. inter-biraderi unions) has a strong effect on recessive genetic disease burden. This will help explain to readers how fine-scale population structure has real-world implications for human health and disease. Other minor comments are described below.

Lines 66-67: This description reflects the ancestry of most but not all groups living in South Asia. For example, Tibeto-Burman speakers in the Northeast of the subcontinent are historical migrants from Southeast Asia, while Austro-Asiatic speakers are hypothesized to be the autochthonous inhabitants. See, e.g., ref. 58 [Basu et al.] for more details.

Line 128: Should be "markers"

Lines 137-140: Maybe it's just me, but I couldn't tell from the color scheme whether Choudhry is a subset of Jatt, especially in Figure 1b.

Lines 203-205: An alternative explanation is that the age of the biraderi groups is not old enough to expect such Y-haplotype stratification.

Lines 224-229: I don't know much about the NeON package (or the underlying McEvoy paper) but I'm a bit skeptical of any method's ability to estimate very recent split times (e.g., 10's of generations) because of sensitivity to model assumptions. I'm not sure I have a specific recommendation, but as far as I know there are no simulation results showing the approach is appropriate for such recent time scales.

Lines 235-240: Greater 'divergence' does not necessarily reflect an older divergence time, and could potentially be explained by greater genetic drift caused by small population size.

Figure 3b: I'm a little surprised that these curves do not increase more in the last 10 generations given what we know about the increased census size over the past several hundred years. Does this potentially reflect changes in endogamy/consanguinity in the past 10 generations?

Lines 298-301: This is an interesting observation and I think it might be worthwhile to have some discussion of the implications (especially of the higher F_{ROH} for individuals with one parent born in the UK and one born in Pakistan).

Lines 309-310: Are there details somewhere of the specific methodology used to estimate 92% accuracy?

Figure 5a: Is there a pink line missing?

Line 434: I would replace "Interestingly" with another word. The observation is expected if current and past intra-biraderi marriage rates are correlated.

Has anyone looked at disease prevalence stratified by biraderi? If not, it might be helpful to state this as a future goal in the Discussion.

Lines 664-665: Any particular reason why phase 3 data were used over the high-coverage 1KGP data generated by the New York Genome Center. There is also the GenomeAsia pilot project reference panel that contains a larger number of South Asian individuals.

Line 720: Should this be ">" instead?

Lines 785-786: It seems more appropriate to use genetic distances rather than physical ones. Please justify why the latter was used.

Reviewer #3 (Remarks to the Author):

Fine-scale population structure and demographic history of British Pakistanis

The authors have assembled genome-wide SNP and exome dataset of >4000 British Pakistani individuals to study demographic inference and impact of cultural practices like endogamy and consanguinity. The analysis performed is compelling but I have some concerns about technical details that I discuss below.

1. Comparison of exome, SNP chip and genome-wide data

The study uses data from exome and SNP chip data from individuals in the Born in Bradford project. The authors do not describe if they are using off-target SNPs in the exome or using coding variants for all the analysis. The coding regions are generally conserved across individuals and so its not clear if they give an unbiased picture of genetic diversity in the population. It would be useful if the authors can do some in-silico analysis with 1000 Genomes or other publicly available datasets to show that the inferences using exome would be reliable.

2. Sampling scheme

Are the four grandparents of each individual from the same biraderi thus is structure reflects the ancestral structure in Pakistan or are these individuals more mixed than groups in Pakistan? Would be worth describing the sampling scheme.

3. Comparison with groups outside the data cohort

The patterns shown in figures 1 & 2 are interesting though hard to interpret since the authors do not include populations from outside the BiB cohort. Supp. Figure 3-4 include some figures but still make it hard to understand the structure in BiB relative to worldwide pops. Key populations like West Eurasians, Onge and other reference populations should be included in PCA, Admixture, TreeMix and other analysis. How do the clusters seen in ADMIXTURE and fine-structure analysis relate to South Asian ancestral populations? What is the choice for K=4 based on?

4. Comparison with previous models and studies

Previous analysis have shown that the model of ANI and ASI explains the genetic variation in India, and in some populations from Pakistan. This is the largest dataset of Pakistanis and it would be useful to assess if this model of ANI and ASI admixture provide a reliable fit to the overall data in Pakistanis or are there other ancestries present in Pakistani individuals? It would also be useful to comment on how the BiB individuals compare to the Sindhi and Pathan samples in HGDP (sampled from Pakistan). Moreover, the Y chr analysis should be in context with what is known about the haplogroups in the ancestral populations of South Asians, namely Steppe Pastoralist and Iranian farmers. Estimates of divergence times seem very recent in Fig 2 and not clear if these account for the history of admixture in the group?

5. IBD / ROH analysis

One concern about these results is that they are based on exome data. Exomes are more conserved and have lower Ne and hence some confounding is expected due to that. Further, figure 3/ 4 shows results of IBDscore and IBDNe in Europeans which looks very significantly different in non-Pakistani groups. Comparison of exome vs. non-exome would be useful here and especially when comparisons are made its important to match the source of data so using exome data in 1000G. It is also important if the authors can discuss the role of phasing errors impacting their results – some of the population sizes are

super large ($N_e = 1M$). Phasing can be particularly challenging in non-European samples since reference data available are limited, though if families were used there could be significant robustness.

Response to reviewers

Please note that substantial changes to the manuscript have been made in blue text.

Reviewer #1 Report

This manuscript describes analyses of British Pakistanis and particularly presents an investigation of fine-scale population structure and history in Pakistanis. The data and results are of interest. However, I do have some major concerns which I hope the authors can address in a revised version.

(1) It is challenging to conduct traditional population genetic studies in a population with high consanguinity and endogamy because random sampling of “un-related” individuals is not guaranteed. Unfortunately, however, most statistical methods applied in this study assume variables to be independent. Analyses at a “fine-scale” are especially affected and many results&conclusions could be invalidated. For instance, clustering analysis, estimation of divergence times and historical effective population size changes, etc.

We agree with the reviewer that identifying unrelated individuals is a challenge in populations with high endogamy and consanguinity, and that this poses a problem for population genetic analyses. For this reason, we tested several different relatedness estimators (PLINK’s \hat{r} , and the PropIBD and kinship estimators from KING) and chose to use the one that identified the most relative pairs (KING’s PropIBD; see Supplementary Figure 2). We then removed one of each pair of individuals identified as third degree relatives or closer by PropIBD. We chose to use this conservative approach in identifying unrelated individuals precisely because we were concerned that our population genetic inferences would be affected if we had employed the more lenient, commonly used PLINK estimator; specifically, we were concerned that we would overestimate the degree of endogamy and divergence times, and underestimate N_e .

As an alternative to KING, we also tried the recently released PONDEROSA method to estimate relatedness (Williams et al., biorxiv, 2020) which was designed to work in endogamous populations. However, we found that it did not work well in our sample because its identification of third-degree and more distant relatives relies on connecting people up via closer relationships, which requires sufficient numbers of pairs who are first- and second-degree relatives for training, which we did not have.

The original version of our paper contained several analyses that confirm the robustness of our relatedness estimates, and we have supplemented them with an additional analysis.

1. Supplementary Figure 2c illustrates that the KING PropIBD cutoff we have used identifies the vast majority of spousal pairs who self-declared as first cousins. We note that first-cousin relationships may actually be over-reported, since the difference between the various types of cousins is reportedly not well understood in the community.
2. Supplementary Figure 14e also suggests that the KING PropIBD estimate is working well at capturing third-degree relatives and closer. For all of the children who were inferred by our ROH-based consanguinity algorithm to have parents who were first cousins or closer (indicated in blue in the figure), the parents were inferred by KING PropIBD (a completely independent approach) to be third- or second-degree relatives.

3. New analysis: To further confirm that the KING PropIBD metric is doing an adequate job of identifying third-degree relatives and closer, we simulated matings between individuals from the same self-declared sub-group and generated relatives of different types. We then used the KING PropIBD metric to estimate relatedness between these simulated relatives. The results suggested that this metric was able to estimate the true relatedness accurately for third degree relatives and closer (Supplementary Figure 2e). More detail is now given in the Methods (lines 633-642).

Finally, we conducted another new analysis to confirm that the fineSTRUCTURE clustering in Figure 2a, which defined the homogeneous groups used in subsequent analyses, was not affected by a failure to remove relatives. We randomly downsampled to 60% of each cluster from Figure 2a four times and ran fineSTRUCTURE for each downsampled dataset. As shown in Supplementary Figure 20, our clustering approach was robust even across downsampled replicates. Although the topology of the tree changed somewhat (as expected since the tree is not rooted in the hierarchical clustering), all homogeneous groups were recapitulated. This suggests that the structure seen in Figure 2a was unlikely to be biased by the accidental inclusion of relatives. This has been added in as a new Supplementary Note (Robustness of clustering approach using fineSTRUCTURE).

In summary, multiple lines of evidence suggest we have properly removed relatives for our population genetics analyses. In fact, we have probably been more conservative than similar papers that have studied population structure and history in endogamous/isolated populations (e.g. Belbin et al., Cell, 2021, or Pathak et al., AJHG, 2018). As a result, we may have been over-cautious and removed some individuals who are not, in fact, third-degree relatives. We now comment on this in the Discussion section (lines 244-248).

(2) In admixture analysis, was $K=4$ the best-supported value for K , based on the cross-validation error? If not, this would indicate potential substructure within the populations that could be of interest, as some more clusters were presented later using other analyses.

Yes, this was the best K according to cross-validation error. This information was reported in the Methods section: "ADMIXTURE v1.3 was run on the pruned datasets, and the cross validation (CV) error was calculated for identifying the best K value, which was found to be 4 for Figure 1c and 8 for Supplementary Figure 4a."

(3) In the Abstract, the description "Here, we investigate fine-scale population structure, history and consanguinity patterns using genetic data from >4,000 British Pakistanis." is not informative. The authors should make it very clear that they analyzed "genotype array data and exome sequence data", and the exact sample size for each type of the data.

(4) It turned out that the maximal sample size of unrelated subjects, assuming the authors' applied correct analysis, was 2,200. For example, according to the main text, "Most analyses in this paper are based on 2,200 unrelated mothers genotyped on the CoreExome array (~251,853 SNPs with minor allele frequency >1% post-quality control), with 1,616 unrelated children (CoreExome) and 228 unrelated fathers (GSA) used in some analyses." Therefore, the ">4000" in the Abstract is an extremely exaggerating number, the authors should not overstate the effective sample size (2,200) for real analysis, which is otherwise misleading.

(5) It does not make sense that the authors “assembled a large dataset of 7,180 individuals” for the purpose of this study. The Illumina CoreExome array of 2,200 “un-related” mothers are the only useful data for this paper. Therefore, I would suggest the authors use only these 2,200 or effectively 1,520 “un-related” mothers for this study.

It is true that most analyses were based on either the 2,200 unrelated mothers on the CoreExome or a subset of these (N=1,520) who fell into the sixteen self-reported subgroups with >20 individuals. However, the analysis of consanguinity patterns mentioned in the abstract also included analysis of 1,616 unrelated children (Supplementary Figure 13 and 14). Nonetheless, we have changed the abstract to mention the 2,200 number at the reviewer’s request. We have not mentioned the sample size for the exome sequence data in the abstract since this was only used for the result in Figure 6, which is not mentioned there, and the word count is limited. We have added Supplementary Table 18 to show the sample sizes and data sources used for each display item.

Moreover, another advantage of this data set would be to possibly reduce the potential batch effects stemmed from a combination of data generated by different technologies or platforms.

The vast majority of analyses in the paper are conducted only using data from the CoreExome chip. The Y chromosome analyses were on the GSA chip alone, so batch effects will not be a problem here. The only analyses for which we have combined data from different platforms is that in Supplementary Figure 2cd. For this, we had to merge the data from the mothers on the Illumina CoreExome chip with data from the fathers on the Illumina GSA chip (since no fathers were genotyped on the CoreExome chip). We don’t anticipate that batch effects will have had a major effect on this analysis as we had already restricted it to SNPs with <1% missingness and passing the HWE filter on each chip (see Methods section “Quality control of genotype chip data”). Besides, this analysis was a very minor part of the paper, included only as a quality control check of our relatedness estimation.

(6) Taken together, I think the authors should better claim a sample size of 2,200 or 1,520 for this paper. The following description in the text is misleading. “We assembled a large dataset of 7,180 individuals with Pakistani ancestry (Supplementary Tables 1 and 2) from the BiB project, of which 5,669 had been genotyped on the Illumina CoreExome array and 1,511 on the Illumina Global Screening Array (GSA); 2,484 of these also had exome-sequence data.”

We have added additional sentences to that paragraph (lines 112-118), which we hope clarify what sample sizes were used where: “Most analyses in this paper are based on genotype data from 2,200 unrelated mothers (CoreExome array; 251,853 SNPs with minor allele frequency >1% post-quality control). Some analyses used genotype data from 1,616 unrelated children (CoreExome; Supplementary Figures 13 and 14) or 228 unrelated fathers (GSA, Supplementary Figure 7 and Supplementary Table 6), and Figure 6 used exome-sequence data from 2,484 mothers. Supplementary Table 18 indicates which samples were used in which analyses.”

Because some analyses based on a small number of males might be not reliable, or the conclusion might be misleading.

We agree that the sample size of males used for the Y chromosome analysis is small. However, we only attempted to draw a very limited conclusion from this analysis, which was that males from the same biraderi group did not tend to cluster together by Y haplotype. Visual inspection of Supplementary Figure 7 suggests that this conclusion is not likely to qualitatively change with a larger sample size. However, we have added in a mention of this small sample size into the Results (line 205): “Although 79% of the individuals carried the IJ haplogroup, and the sample size is relatively small, we did not find the clear delineation of some groups observed in analysis of the autosomal data.” Furthermore, the Discussion (line 532-535) acknowledges that a larger sample size would be desirable: “For example, whole-genome sequence data on a larger sample size of males would allow us to leverage more Y chromosomal markers to further explore whether biraderi membership has indeed been passed down patrilineally in recent time.”

(7) Sample size can be a key factor for comparative analyses. For example, the following results need some further analysis with random sampling individuals of similar sample size. “However, the self-reporting seemed to be less reliable for the individuals who said their parents were first cousins once removed or second cousins (Figure 4b). Overall, consanguinity was under-reported in the mothers and slightly over-reported in the children, with 78% of the mothers inferred to have parents who are second cousins or closer compared to the 57% reported, versus 58% inferred and 63% reported for the children (Supplementary Figure 14).”

Rather than random sampling, we have conducted some formal statistical tests using z-tests for a difference in proportions. These are now mentioned on lines 330-336 and 837-841. We apologise for this omission in the original manuscript.

(8) As the authors were also aware, one limitation of the study is a small sample size that would not allow reliable group-level inference about the demographic history of founder events. However, uneven/non-random sampling of the individual's under-representing diversity might be more problematic in population genetic studies. The authors should take caution in interpreting their results.

We believe this is a reasonably random sample of British Pakistanis living in Bradford, at least amongst the population of mothers giving birth there in 2007-2010, since all such individuals were invited to participate and 80% agreed (see Methods, line 526). However, we acknowledge in the Discussion that this is clearly not a random sample of all Pakistanis in Pakistan and thus that our findings may not be generalisable to all Pakistanis (lines 526-529).

Reviewer #2 Report

This manuscript provides a detailed and novel analysis of fine-scale population structure, endogamy and consanguinity among British Pakistanis in the BiB study. Overall, I think the work will interest many readers of Nature Communications, though I think there are some changes I think the authors could make to more directly highlight the relevance of their work. The most important of these is to explicitly discuss in the abstract the results of Figure 6, which show that endogamy (as quantified by intra vs. inter-biraderi unions) has a strong effect on recessive genetic disease burden. This will help

explain to readers how fine-scale population structure has real-world implications for human health and disease.

We thank the reviewer for these kind comments.

We appreciate the point about mentioning the implications for recessive disease burden in the Abstract. This finding may be sensitive with the Pakistani community, and has the potential to attract unwanted negative attention from certain commentators who may not read beyond the Abstract, where there will not be space to include necessary caveats. Hence, we made a conscious decision not to include this in the Abstract. We have prepared a set of Frequently Asked Questions and responses for lay people about the study, which attempts to present these findings in a sensitive way, with these appropriate caveats.

Other minor comments are described below.

Lines 66-67: This description reflects the ancestry of most but not all groups living in South Asia. For example, Tibeto-Burman speakers in the Northeast of the subcontinent are historical migrants from Southeast Asia, while Austro-Asiatic speakers are hypothesized to be the autochthonous inhabitants. See, e.g., ref. 58 [Basu et al.] for more details.

We have rephrased the statement to highlight that most but not all South Asians derived their ancestry from the ANI and ASI components (line 65). We have also cited Basu et al. accordingly (reference 20).

Line 128: Should be "markers"

We thank the reviewer for spotting this typo, which we have now corrected (line 131).

Lines 137-140: Maybe it's just me, but I couldn't tell from the color scheme whether Choudhry is a subset of Jatt, especially in Figure 1b.

This is more evident in Figure 1a, where Choudhry and Jatt come out on PC1. Given the large number of groups, it is difficult to choose a colour scheme with no ambiguity, and since none of the other reviewers have commented on this, we propose to leave it as is, but we have noted the colours in the main text (line 139): "The fact that the Choudhry (dark orange colour) and Jatt (teal colour) subgroups cluster together...".

Lines 203-205: An alternative explanation is that the age of the biraderi groups is not old enough to expect such Y-haplotype stratification.

We thank the reviewer for this suggestion and we think it is a plausible explanation. We have modified the end of that paragraph to say (lines 208-212), "This is consistent with previous findings in Punjabi Rajputs and in Jatts, and may suggest at least three possible explanations which are not mutually exclusive: the founders of each biraderi may have included males with several different haplogroups, the age of the biraderi groups may not be old enough to lead to strong Y-haplotype

stratification, and/or historically the patrilineality of the biraderi system may not have been very strict.”

Lines 224-229: I don't know much about the NeON package (or the underlying McEvoy paper) but I'm a bit skeptical of any method's ability to estimate very recent split times (e.g., 10's of generations) because of sensitivity to model assumptions. I'm not sure I have a specific recommendation, but as far as I know there are no simulation results showing the approach is appropriate for such recent time scales.

In the NeON package paper, the authors performed simulations using the python library simuPOP to show the robustness of inferred divergence times under different demographic scenarios. These results showed that under all scenarios they tested, recent divergence times (from 100 to 1000 generations) were reliably inferred and that confidence intervals were small. However, the divergence times inferred in our manuscript are more recent than these, so to confirm that our inferences were robust, we did forward simulations using simuPOP. These are summarised in a new Supplementary Note (Assessing accuracy of NeON divergence time estimates), and the results shown in Supplementary Figure 21. They suggest to us that NeON is inferring the split times accurately.

Lines 235-240: Greater 'divergence' does not necessarily reflect an older divergence time, and could potentially be explained by greater genetic drift caused by small population size.

We agree with the reviewer on this point, in general, but we believe that our simulation results demonstrate that the divergence time estimates are fairly accurate in this case, as described in response to the previous comment. However, we changed the text accordingly to include this possible explanation (lines 244-248).

Figure 3b: I'm a little surprised that these curves do not increase more in the last 10 generations given what we know about the increased census size over the past several hundred years. Does this potentially reflect changes in endogamy/consanguinity in the past 10 generations?

This finding may well reflect increased consanguinity and endogamy in more recent times, possibly even as a result of the migration to the UK. To our knowledge, there is no empirical quantitative historical data on this, but as we note in the manuscript, “Historical records suggest that endogamous practices became even stricter during the colonial times of the 19th century, as the social classification system was reinforced by the British to solidify their political authority, enable rationalised taxation, and establish rules about property.”

Lines 298-301: This is an interesting observation and I think it might be worthwhile to have some discussion of the implications (especially of the higher F_{ROH} for individuals with one parent born in the UK and one born in Pakistan).

Indeed, it is interesting. We have added the following sentence to the manuscript (lines 311-315): “This fits with some prior evidence that cousin unions may be preferred for trans-national marriages (Shaw, 2014) among first- and second-generation British Pakistanis, for whom they can be a means

of ensuring 'cultural continuity' and socio-economic support, and to allow siblings separated by migration to reconnect through the marriages of their children."

Lines 309-310: Are there details somewhere of the specific methodology used to estimate 92% accuracy?

The accuracy is determined by simulating an equal number of individuals from each of the model classes and assessing what fraction of individuals are classified correctly by the neural net. This is now clarified in the Methods (lines 830-835): "We tested the algorithm by randomly sampling individuals ten times, each time simulating an equal number of individuals from each of the ten model classes. Each of the ten times, we assigned an individual to a parental relatedness category based on the category for which they got the most classifications across the ten trials. The accuracy was determined as the fraction of individuals classified correctly by the neural net."

Figure 5a: Is there a pink line missing?

The pink line is beneath the green one. We have added a sentence on this to the legend.

Line 434: I would replace "Interestingly" with another word. The observation is expected if current and past intra-biraderi marriage rates are correlated.

We thank the reviewer for this suggestion. We replaced "Interestingly" with "As one might expect" (line 457).

Has anyone looked at disease prevalence stratified by biraderi? If not, it might be helpful to state this as a future goal in the Discussion.

To our knowledge, this has not been done, but indeed would be interesting. We have added to the Discussion (line 509-514), "Our findings suggest that clinicians should consider recording parents' biraderi groups as well as close relatedness in genetic consultations..... Recording *biraderi* information would enable further research into prevalence of different diseases in different biraderi groups, the impacts of endogamy and the possible presence of disease-causing founder mutations."

Lines 664-665: Any particular reason why phase 3 data were used over the high-coverage 1KGP data generated by the New York Genome Center. There is also the GenomeAsia pilot project reference panel that contains a larger number of South Asian individuals.

The high-coverage 1KP and GenomeAsia reference panels were not available at the time we completed this work. However, we doubt that slight changes in phasing that might come from using a different panel would substantially impact our main conclusions from fineSTRUCTURE. This is because, as we describe below in response to the final question from reviewer 3, similar clustering is obtained using an IBD-based method that does not rely on phased data.

Line 720: Should this be ">" instead?

Indeed, apologies for this typo, which we have now fixed (line 772).

Lines 785-786: It seems more appropriate to use genetic distances rather than physical ones. Please justify why the latter was used.

When considering the impact of homozygosity on phenotypes, the ROHGen consortium used physical distances presumably because they better capture the number of potentially functional base pairs in the genome that are affected by the ROHs (Clark et al., Nat Comms, 2019). In practice, we see that F_{ROH} calculated with physical distance is highly correlated with F_{ROH} calculated with genetic distance, so we propose to leave the figures as they are. The following figure illustrates this. The blue line of best fit is $y=0.97x + 0.002$, and the correlation is 0.99.

Reviewer #3 Report:

The authors have assembled genome-wide SNP and exome dataset of >4000 British Pakistani individuals to study demographic inference and impact of cultural practices like endogamy and consanguinity. The analysis performed is compelling but I have some concerns about technical details that I discuss below.

1. Comparison of exome, SNP chip and genome-wide data

The study uses data from exome and SNP chip data from individuals in the Born in Bradford project. The authors do not describe if they are using off-target SNPs in the exome or using coding variants for all the analysis. The coding regions are generally conserved across individuals and so its not clear if they give an unbiased picture of genetic diversity in the population. It would be useful if the authors can do some in-silico analysis with 1000 Genomes or other publicly available datasets to show that the inferences using exome would be reliable.

Apologies if this was not clear in the manuscript. The exome sequence data were only used for ascertaining the potentially pathogenic variants in Figure 6 of the paper and were not used for any inferences about population structure, demographic history or consanguinity rates. All the other analyses are based on the SNP chip data, specifically the common SNPs which primarily come from the “Core” (imputation backbone) part of the CoreExome chip rather than the rare exonic variants. We have clarified this in the first paragraph of the Results (lines 112-118), and added Supplementary Table 18 which lists all the datasets and sample sizes used for each analysis.

2. Sampling scheme

Are the four grandparents of each individual from the same *biraderi* thus is structure reflects the ancestral structure in Pakistan or are these individuals more mixed than groups in Pakistan? Would be worth describing the sampling scheme.

The sampling scheme for Born in Bradford simply involved inviting all women (of any ethnicity) giving birth in the Bradford Royal Infirmary in 2007-2010 to participate in the study, and ~80% of those eligible accepted. We have added a sentence into the first paragraph of the Methods about the recruitment (lines 562-563). Sampling was agnostic to individuals' *biraderi* group and to their ancestors' *biraderi* group. The marriage patterns of the Pakistani mothers and their parents with respect to *biraderi* were described in a previous paper (Small *et al.*, Journal of Biosocial Science, 2016). We are not aware of any comparable data from Pakistan so we do not know the extent to which this structure reflects the ancestral structure there.

3. Comparison with groups outside the data cohort

The patterns shown in figures 1 & 2 are interesting though hard to interpret since the authors do not include populations from outside the BiB cohort. Supp. Figure 3-4 include some figures but still make it hard to understand the structure in BiB relative to worldwide pops. Key populations like West Eurasians, Onge and other reference populations should be included in PCA, Admixture, TreeMix and other analysis.

We decided not to include results about the genetic relationship of the BiB Pakistanis with external populations in the main text or figures because we thought these results were less interesting. However, the first Supplementary Note covers this, together with Supplementary Figures 3 and 4 and Supplementary Tables 4, 5 and 15. The overall conclusion was that the BiB Pakistanis have a similar genetic profile to the HGDP Pathans and Sindhi and display little variation in their ancestral components in this broad context. Key worldwide populations including west Eurasians and Onge were already included in the ADMIXTURE analysis in Supplementary Figure 4a. We improved the ADMIXTURE plot to better show the patterns of genetic sharing between the BiB subgroups and other worldwide populations. We have also added a PCA plot that includes Europeans, Central Asians and South Asians, indicating key populations (Supplementary Figure 3b).

Finally, we computed outgroup f_3 -statistics with modern populations on each BiB subgroup separately, since Supplementary Figure 4c shows the results with them all combined. The results are below and show very similar genetic affinity between the different BiB subgroups and the external reference populations tested, as we observed with ancient genomes in Supplementary Figure 4b. We

added these results to Supplementary Table 4 and expanded the text in the Supplementary Note to better explain the structure of BiB Pakistanis relative to other worldwide populations.

How do the clusters seen in ADMIXTURE and fine-structure analysis relate to South Asian ancestral populations?

Our ADMIXTURE results (Supplementary Figure 4a) indicated that the various BiB subgroups have a similar genetic profile when compared to other worldwide populations, with the possible exception of the Pathans, as noted in the Supplementary Note. Thus, we did not specifically look at sharing between the fineSTRUCTURE-defined clusters and South Asian ancestral populations, since we don't expect this conclusion to change. In the outgroup f_3 -statistics results in Supplementary Table 4 we have included representative populations that are proxies for the ANI (Uttar Pradesh Brahmins, Kalash, Sindhi, Pathan) and ASI (Onge, Jarawa, Irlula) ancestral populations. As expected, the BiB subgroups have higher genetic sharing with populations deriving most of their ancestry from the ANI component (Basu et al., PNAS, 2016).

These results are consistent with the proportions reported in previous publications, including the estimates in Narasimhan *et al.* (Science, 2019) modelling present South Asians (including Rajput) using ancient samples.

Overall, all the BiB subgroups have similar sharing with all the populations tested, implying they all come from the same ancestral population (although the Pathan have slightly more ANI ancestry, consistent with their originating from further to the northwest).

What is the choice for K=4 based on?

This was the best K according to cross-validation error. This information was reported in the Methods section: “ADMIXTURE v1.3⁶⁹ was run on the pruned datasets, and the cross validation (CV) error was calculated for identifying the best K value, which was found to be 4 for Figure 1c and 8 for Supplementary Figure 4a.”

4. Comparison with previous models and studies

Previous analysis have shown that the model of ANI and ASI explains the genetic variation in India, and in some populations from Pakistan. This is the largest dataset of Pakistanis and it would be useful to assess if this model of ANI and ASI admixture provide a reliable fit to the overall data in Pakistanis or are there other ancestries present in Pakistani individuals?

Following Narasimhan *et al.* (Science 2019), we used qpgraph to confirm the ANI and ASI model explains the genetic variation seen in the Bradford subgroups. Narasimhan *et al.* already tested the Indian Rajputs and other Pakistani populations in their paper. We used the same qpgraph parameters and populations as in Narasimhan *et al.* to test the model for each of the Bradford subgroups. We found that all Bradford subgroups fit the ANI and ASI model, with all subgroups having the majority of their ancestry derived from the ANI component. The largest deviation between theoretical and empirical f -statistics was Z-score = 2.9, suggesting a good fit of the model considering the vast number of f -statistics analysed. Pathan had the highest proportion of the ANI component (80%) and Qasabi the lowest (61%). We added these new results in the Supplementary Note (lines 1112-1119), Supplementary Table 15 and Supplementary Figure 19 of the manuscript, with Methods in lines 695-702.

A further exploration of deep demographic history is beyond the scope of this paper, and ideally would be conducted using a dataset that contained Pakistanis from additional ethnic groups other than Punjabi and Kashmiri.

It would also be useful to comment on how the BiB individuals compare to the Sindhi and Pathan samples in HGDP (sampled from Pakistan).

HGDP Pathan and Sindhi samples are included and labelled in our plots (Supplementary Note, Supplementary Figures 3c,d and 4a,c). We have now clearly emphasised these populations in the improved ADMIXTURE plot (Supplementary Figure 4a). We added these two sentences to the Supplementary Note (lines 1091-1093): “The majority of the BiB samples are most similar to the

Sindhi, out of the reference populations included. Only the BiB Pathans stand out; they have a higher fraction of the pink and blue components seen in Europeans, as seen in the HGDP Pathans.”

Moreover, the Y chr analysis should be in context with what is known about the haplogroups in the ancestral populations of South Asians, namely Steppe Pastoralist and Iranian farmers.

Most of the Bradford subgroups belong to IJ*, G* and P* haplogroups that are common in modern European, Central and South Asia populations (Hallast *et al.*, Human Genetics, 2021). We used aYChr-DB (Freeman *et al.*, NAR Genomics and Bioinformatics, 2020) to look for reported Y-chromosome haplogroups in ancient samples. The IJ*, G* and P* haplogroups have been found in West and Central Asia individuals from the Neolithic period onwards, including Iranian farmers and one sample from the Central Steppe, two of the major ancestral contributions to modern-day South Asians (Narasimhan *et al.*, Science, 2019). More Y chromosome SNPs would be required to give better resolution of the haplogroups. We added this part to the Supplementary Note of the manuscript (lines 1130-1135).

Estimates of divergence times seem very recent in Fig 2 and not clear if these account for the history of admixture in the group?

As with many (if not all) methods that infer divergence time, Neon does not specifically account for admixture, but if there has been admixture after the initial divergence, this will make the divergence time estimates more recent than the point at which the populations initially started to separate. In the case of Neon, this is because the admixture reduces the F_{ST} .

We noted this in the Results (lines 237-241): “The estimates in Figure 3a suggest that the history of these subgroups cannot be considered as a series of clean splits between ancestral populations; rather, it appears that several groups began to differentiate from one another around the same time, with some degree of admixture persisting between the groups after their initial divergence.” It is also notable that the Kashmiri have the most recent split times with other groups and were also inferred to have ADMIXTURE from several of those groups with TreeMix.

5. IBD / ROH analysis

One concern about these results is that they are based on exome data. Exomes are more conserved and have lower N_e and hence some confounding is expected due to that. Further, figure 3/ 4 shows results of IBDscore and IBDNe in Europeans which looks very significantly different in non-Pakistani groups. Comparison of exome vs. non-exome would be useful here and especially when comparisons are made its important to match the source of data so using exome data in 1000G.

Sorry if this was not clear. As mentioned above, the exome sequence data were only used for ascertaining the variants in Figure 6 of the paper. For calling IBD regions and ROHs, we used ~250k genome-wide common SNPs from the CoreExome chip.

It is also important if the authors can discuss the role of phasing errors impacting their results – some of the population sizes are super large ($N_e = 1M$). Phasing can be particularly challenging in

non-European samples since reference data available are limited, though if families were used there could be significant robustness.

In Supplementary Figure 11b, we show a comparison of $IBDN_e$ results using GERMLINE (phased data) versus IBDseq (unphased data), and indeed GERMLINE gives higher $N_e$ estimates for most populations, which is probably due to phasing errors (as previously noted in the Supplementary Note, now lines 1166-1167). For this reason, and as recommended by the original $IBDN_e$ paper, we used IBDseq to generate the main $IBDN_e$ results shown in Figure 3b as well as those in Supplementary Figure 10 and Supplementary Figure 11ac.

Only three of our main analyses made use of phased data: fineSTRUCTURE (Figure 2a), the IBD scores (Figure 4a; using GERMLINE calls), and the generation of gametes in the simulations of pedigrees to infer consanguinity (Figure 5).

In theory, there is certainly the potential for phasing errors to impact the fineSTRUCTURE results. Our main use of the fineSTRUCTURE results was to define homogeneous subgroups using a combination of the fineSTRUCTURE inference and the self-reported biraderi information. To check whether this classification was robust to phasing errors, we applied an orthogonal IBD-based method to explore fine-scale structure, specifically that used by Dai *et al.* (AJHG, 2020). We called IBD segments from unphased data with IBDseq, constructed a matrix of total IBD sharing between individuals, and then identified clusters of individuals using Louvain clustering in the igraph package in R. The following figure shows the fraction of individuals in each fineSTRUCTURE cluster that are placed into the indicated igraph clusters. This figure indicates that the homogeneous sub-groups inferred using a combination of fineSTRUCTURE and the self-reported biraderi labels are largely recapitulated using this other IBD-based approach which uses unphased data. Hence, we do believe that phasing errors in fineSTRUCTURE are not likely to have had much of an impact on the key results of the paper.

We also used phased data for running GERMLINE which we used to determine IBD scores in Figure 4a and Supplementary Figure 12. We have now calculated the IBD scores using IBDseq data (unphased) and see that the results are still very consistent (Supplementary Figure 12e).

Finally, phased data were used in the generation of gametes when simulating pedigrees to infer consanguinity based on patterns of ROHs (Figure 5, Supplementary Figure 14). The phased haplotypes were simply used to simulate offspring with differing degrees of consanguinity, in which ROHs were inferred, and it is the distribution of these ROH lengths that is the basis of our inference. We do not need to phase the individuals whose level of parental relatedness we are trying to infer; we simply compare their ROH distribution to that of individuals in the simulated data. Thus phasing errors should not impact this analysis since the actual haplotypes that are homozygous do not matter, and the ascertainment of the ROHs does not rely on phased data.

REVIEWERS' COMMENTS

Reviewer #1 (Remarks to the Author):

I appreciate the authors made a lot of effort in revising the manuscript. The manuscript seems to have been considerably improved. I understand there are some concerns that are difficult to address with the current data. I would support publishing this manuscript.

Reviewer #2 (Remarks to the Author):

The authors have addressed my previous concerns and I think the revision is a substantial improvement over the original submission.

Reviewer #3 (Remarks to the Author):

The authors have addressed most of the technical concerns raised. The manuscript is much improved and interpretation of results is clearer. For a couple of results, it would be useful to still add clarification about the numbers:

1. Split times

While most (if not all) available methods do not account for admixture while estimating split times, depending on the substructure in the population, the results will be more or less biased. In this population, the admixture rates are between 60-80% and thus not accounting for admixture is very problematic. I would clarify this in the results and discussion.

2. IBDNe results in Figure 3b

Present-day effective population sizes in some groups are between 100k-10M -- this is most likely due to phasing errors and this should be clearly stated in the main text so the readers can interpret these results. Further, this population is admixed and the vanilla model in IBDNe is not appropriate as the gene flow can also impact the population size distribution. These caveats should be included in the results.

Reviewer #4 (commenting specifically on the FAQ):

General

Really interesting to read this and consider if it could be tweaked. I am not qualified to review the main paper, although I am able to appreciate how much work has gone into it and that its findings give major

insights into population histories. I hope that my few comments on the A&Q supplement are helpful. Generally, it strikes me as thoroughly appropriate, reflecting the major findings and unlikely to cause confusion. However, I'd like to raise one thought:

It seems that you have decided not to say much in this Q&A supplement about the comparisons you have made with other studies of genetic structure in Central and South Asian (including Indian) groups. I think it would be good to look at this decision again, if indeed it was a specific decision. And perhaps you could even say something about the comparable studies of other groups with a lot of their own, specific recessive diseases. The obvious, other (non-South Asian) comparison group would be the Ashkenazi Jewish populations, where the explanation for the high incidence of recessive diseases could have included endogamy and sometimes consanguinity but may also include marked fluctuation in the size of population groups from persecution. What about a mention of at least some of these other studies?

Typos or etc:

Line 108: "people tended to marry within one's own biraderi": I would suggest changing "one's" => "their"

Lines 160-162: The phrase "a particular DNA variant" is not quite accurate in the broader context of recessive disease. I think it may be better to replace it with something like, "variants that both impact the function of the same gene" as this may avoid the generation of misunderstandings about recessive disease always being the result of common ancestry leading to alleles that are IBD.

Angus Clarke

Reviewer #5 (commenting specifically on the FAQ):

I have read both files and I am sending you a list of specific comments on the supplementary file plus some notes for the main one because some of my comments also have implications for the manuscript; the manuscript has already been peer-reviewed, so my suggestions are to improve clarity and to ensure consistency in language use across the two documents.

It seems to me there are no particularly contentious statements here. The potential sensitivities concern: debate over whether or not people 'should' marry cousins, the stigmatisation of cousin marriages on grounds of genetic risk, and the sensitivities people might have when asked to state their biraderi identity, given the traditional ranking of biraderis in a status system that has historical links with that of the Hindu caste system. The biraderi system is important politically and economically in Pakistan and it influences marriage choice but does not always determine them. The question of status difference – that is, of potential loss of status or claims to higher status – arises most often when intra (between)- biraderi marriages are discussed. However, this aspect is not directly relevant to this manuscript, and in it the issue of status hierarchy is very much played down.

If the supplementary file aims to address lay readers' "frequently asked questions" - or likely-to-be asked questions – then the authors could slightly expand the section on how 'risky' are cousin marriages and inter-biraderi marriages. Specifically, since the definition of biraderi used in this paper emphasises the patriline, the authors might consider defining 'cousin' from a genetics viewpoint in order to clarify that all first cousins (mother's siblings children, and father's siblings children) are equivalent in terms of shared genetics.

I don't think this would go beyond the assumptions and findings of this research but there is anthropological evidence that in patrilineal societies, people may consider 'father's side' cousins to be genetically closer than those on the mother's side – an idea that does not match Mendelian genetics but that has led some families underestimating risks – e.g. by avoiding father's siblings children as partners for their children in favour of cousins on the mother's side, thinking these are less risky (see e.g. the chapter by Leila Prager, and the chapter by Shaw in *Cousin Marriages: between genetic risk and cultural change*, edited by Alison Shaw and Aviad Raz. Berghahn 2015. Also see Shaw's chapter in a book edited by Veronique Petit et al. *The Anthropological Demography of Health* OUP. 2020).

The supplementary file will be useful for a general readership, and anyone – health professionals and lay people - already familiar with the UK (and wider) debates about 'how risky' cousin marriages are. The findings may also be of interest to social scientists including anthropologists (who these days often view kinship as entirely a social construct!), historians, and scholars of South Asia from all these disciplines.

I have therefore focused on clarifying points or terms used in the supplementary file that did not seem clear or consistent, most especially for non-specialists. These are listed below. [e.g. distinguish singular and plural meanings of biraderi - one biraderi /the biraderi system, vs. biraderis/biraderi groups, to distinguish consanguinity & consanguineous marriages, & check the use of of inter-biraderi & intra-biraderi]. Some of my suggestions implications for the main article as well, for consistency between the documents; these I indicate with a star *, and list briefly at the end.

Specific comments on the supplementary file

Questions 1-3

Lines:

5 – change 'field' to 'fields' (genetics, epidemiology, and anthropology are three separate fields - as far as I can see, only one of the research team is an anthropologist)

*10-11 – the term 'population structure' (in supplementary file & in main article), refers to genetic population structure, and 'history' refers to genetic history. However, for a lay audience these terms will have other meanings. Although there is a glossary, I suggest some rephrasing for clarity/accuracy.

12 – ‘do not translate well to individuals...’ -- perhaps rephrase: ‘may not be the same for people’ of other ethnicities.

*15 – 16 – ‘There is widespread interest in the history, dynamics and structure of populations’ – this is a general and ambiguous sentence that can be deleted. I suggest instead: ‘Research into a population’s genetic history, structure and dynamics can give information that can be compared with information about the populations’ social history, structure and dynamics.

*17-18 Also, a population’s unique social structure and cultural history – such as the biraderi system, and the practice of consanguineous marriage – can influence its genetic structure and genetic variability. (See the Glossary at the end of this paper for definitions of terms used).

Q 3-6

59 ‘In the DNA context’ --- can these 4 words be deleted.

*91 – consanguinity – in my understanding, this term does not refer to a practice - the practice of marrying someone who genetically related (as a second cousin or closer) - but to a quality of a relationship - technically this quality refers to the (quantifiable) extent to which two people have an ancestor in common.

So I think ‘consanguinity’ here should be changed to ‘consanguineous marriage’

92-3 – endogamy – definition - I suggest you delete ‘cultural’ so it reads ‘the practice of marrying...’ reasons: ‘cultural’ does add anything

*105 ‘The Pakistani population’ – does this mean, British Pakistanis? If so, say, the British Pakistani population is made up of people from different ethnic groups, such as...

105-6 ‘The society is organised into kinship networks passed through the male line...’ .

British Pakistanis are not a ‘society’ but a population, and in Pakistan society there are other systems besides the biraderi system. And networks are not passed through the male line but biraderi identity is. So how about rephrasing as follows:

‘Many British Pakistanis identify with a lineage-based system of kinship groups, which we refer to here as the biraderi system’.

*Regarding the use of the word biraderi, to avoid confusing your readers, I suggest that, because you are writing in English, you should follow English grammar - (even though in Urdu/Panjabi you would say ‘one biraderi, two biraderi’). You can either add an ‘s’ (not in italics) to the word when you are referring to two or more biraderis/ discussing biraderis in general, or else rephrase as ‘biraderi groups’.

108, please correct the grammar : - change 'one's' to 'their' and delete 'the' :

'... people tended to marry within their own biraderi, in order to reinforce hereditary social status, occupational identity and land ownership.

110-111 This next sentence can be expressed more simply:

To a certain extent, the biraderi system is still important in Pakistan and among British Pakistanis.

*114 – change consanguinity to consanguineous

Q 7-11

164 – '...(such as a biraderi group)' - and you should italicise the word ' biraderi' for consistency.

167 – in this line, biraderi should be plural to make sense in English. Please either add an 's' or change to biraderi groups.

168 – please change 'have' to 'result in' --- the marriage does not have the child, but the marriage may result in a child...

169 – in this line, biraderi should be plural to make sense in English. Please either add an 's' or change to biraderi groups.

184-7 The sentence in brackets: there should be a plural form for the first to 'biraderi' in this sentence: '.....somewhat accurate for some biraderi groups, since most people who reported coming from those groups were more similar to others in their group than they were to people who reported coming from a different biraderi.

NB again, biraderi should be consistently in italics throughout

Pakistanis 188 – I suggest changing 'the Bradford community' to 'Bradford Pakistanis' (because although Bradford Pakistanis often refer to themselves in this way to distinguish themselves from other ethnic groups, the term implies a commonality and unity that is a bit misleading)

202-4

Could you rephrase, to put more neutrally?,'demonstrate the impact of marriage practices and partner choice on population structure and genomic diversity among Pakistanis'.

210

Could you say, How were British Pakistanis involved in this study? (because although Pakistanis may refer to themselves in this way, 'community' implies a commonality and cultural unity that is somewhat

misleading)

213 can you change 'and on the way findings can be shaped in ways that can improve...' to 'and on how research findings can be used to improve...'

217...advice from people of Pakistani heritage living in Bradford who were....recognised locally as having a particular interest in and knowledge of their local communities, including the historic and contemporary importance of biraderi groups.

220 – could you change 'from the Bradford Pakistani community' to 'born and raised in Bradford' if this is indeed the case?

Glossary – are the terms listed in order of appearance, or in alphabetic order?

*Population structure - This phrase occurs in the title. Could you change this to 'genetic population structure' -(please see below for my comment on demographic history, and on the title) because then the gloss is accurate.

*Demographic history - This term did not come up in the supplementary file, it only occurs in the title of the main article.

Demographic analysis is the statistical analysis of population characteristics such as population size, age, sex, ethnicity, in-migration and out-migration and demographic history refer to changes over time in these rates. Here you are referring to historical changes over time in the genetic structure of populations.

Do you need the word demographic? Could you simplify the title? It would also be useful for lay readers to have the word genetic in the title. A possible reworked title could be:

“Fine-scale genetic population structure and history of British Pakistanis”

*Consanguinity – see comment above about this word in line 91. If you use 'consanguineous marriage' instead, please add a gloss

*Consanguineous marriage – marriage to someone related genetically, as a second cousin or closer

Endogamy – the gloss is fine, but for the example in brackets, it would be helpful to add, 'or from the same biraderi'

*Biraderi - a kinship group with a shared identity (e.g. Jat, or Rajput) within a larger lineage-based

system of kinship groups, which we refer to as the biraderi system.

*Points for main article (page/line numbers give here may not be entirely accurate but worth checking the use of all these terms for consistency throughout the ms):

Population structure – genetic structure? Lines 38, 44, 48-9

Change biraderi to the biraderi system, line 77; -- see above on singular/plural

p. 18/ 562 change to particular biraderi groups

P 3 line 101, also Line 587 – Pakistani community – or British Pakistanis?

p. 2 line 84, add 'the ' – the Gupta empire

Section on broader impact:

line 584, and line 594 – inter-=within; intra= between; also p. 13?

Line 589 – Bradford Pakistani community – or Bradford Pakistanis?

Response to reviewers

Please note that changes to the manuscript have been made in purple text.

Reviewer #1 Report (Remarks to the Autor):

I appreciate the authors made a lot of effort in revising the manuscript. The manuscript seems to have been considerably improved. I understand there are some concerns that are difficult to address with the current data. I would support publishing this manuscript.

We thank the reviewer for the positive feedback about the revised manuscript.

Reviewer #2 (Remarks to the Author):

The authors have addressed my previous concerns and I think the revision is a substantial improvement over the original submission.

We thank the reviewer for the positive feedback about the revised manuscript.

Reviewer #3 (Remarks to the Author):

The authors have addressed most of the technical concerns raised. The manuscript is much improved and interpretation of results is clearer. For a couple of results, it would be useful to still add clarification about the numbers:

We thank the reviewer for the positive feedback about the revised manuscript.

1. Split times

While most (if not all) available methods do not account for admixture while estimating split times, depending on the substructure in the population, the results will be more or less biased. In this population, the admixture rates are between 60-80% and thus not accounting for admixture is very problematic. I would clarify this in the results and discussion.

This has been now clarified in the results and discussion (lines 240-242)

2. IBDNe results in Figure 3b

Present-day effective population sizes in some groups are between 100k-10M -- this is most likely due to phasing errors and this should be clearly stated in the main text so the readers can interpret these results. Further, this population is admixed and the vanilla model in IBDNe is not appropriate as the gene flow can also impact the population size distribution. These caveats should be included in the results.

This has been now added in the results (lines 294-297)

Reviewer #4 (commenting specifically on the FAQ):

General

Really interesting to read this and consider if it could be tweaked. I am not qualified to review the main paper, although I am able to appreciate how much work has gone into it and that its findings give major insights into population histories. I hope that my few comments on the A&Q supplement are helpful. Generally, it strikes me as thoroughly appropriate, reflecting the major findings and unlikely to cause confusion. However, I'd like to raise one thought:

It seems that you have decided not to say much in this Q&A supplement about the comparisons you have made with other studies of genetic structure in Central and South Asian (including Indian) groups. I think it would be good to look at this decision again, if indeed it was a specific decision. And perhaps you could even say something about the comparable studies of other groups with a lot of their own, specific recessive diseases. The obvious, other (non-South Asian) comparison group would be the Ashkenazi Jewish populations, where the explanation for the high incidence of recessive diseases could have included endogamy and sometimes consanguinity but may also include marked fluctuation in the size of population groups from persecution. What about a mention of at least some of these other studies?

First of all, we thank the reviewer for the positive feedback. This is a good point and we agree with the reviewer on the high interest on the topic. We mostly used other modern and ancient human populations to help our understanding of the demographic history of the British Pakistanis. For this manuscript, we did not formally compare our populations to others regarding their incidence of recessive disorders and their potential causes. Therefore, including this piece of research in our manuscript and in the FAQ document would be out of scope. However, we appreciate the importance of mentioning that other human populations might have a higher incidence of recessive disorders due to cultural practices, population bottlenecks and geographical isolation. We added a new point (point 11) in the FAQ document addressing this topic with a few examples of populations that were also mentioned in the manuscript as comparisons.

Typos or etc:

Line 108: "people tended to marry within one's own biraderi": I would suggest changing "one's" => "their"

We thank the reviewer for spotting this error. We have changed it accordingly.

Lines 160-162: The phrase "a particular DNA variant" is not quite accurate in the broader context of recessive disease. I think it may be better to replace it with something like, "variants that both impact the function of the same gene" as this may avoid the generation of misunderstandings about recessive disease always being the result of common ancestry leading to alleles that are IBD.

We thank the reviewer for the comment. This has been replaced in the text.

Angus Clarke

Reviewer #5 (commenting specifically on the FAQ):

I have read both files and I am sending you a list of specific comments on the supplementary file plus some notes for the main one because some of my comments also have implications for the manuscript; the manuscript has already been peer-reviewed, so my suggestions are to improve clarity and to ensure consistency in language use across the two documents.

It seems to me there are no particularly contentious statements here. The potential sensitivities concern: debate over whether or not people 'should' marry cousins, the stigmatisation of cousin marriages on grounds of genetic risk, and the sensitivities people might have when asked to state their biraderi identity, given the traditional ranking of biraderis in a status system that has historical links with that of the Hindu caste system. The biraderi system is important politically and economically in Pakistan and it influences marriage choice but does not always determine them. The question of status difference – that is, of potential loss of status or claims to higher status – arises most often when intra (between)- biraderi marriages are discussed. However, this aspect is not directly relevant to this manuscript, and in it the issue of status hierarchy is very much played down.

We thank the reviewer for the feedback. We would like to clarify that we followed the standard definition of 'intra' and 'inter'. Therefore, we consider the prefix 'intra' for marriages within the same biraderi and 'inter' for marriages between individuals of different biraderi groups. Apologies if this was not clear in the manuscript.

If the supplementary file aims to address lay readers' "frequently asked questions" - or likely-to-be asked questions – then the authors could slightly expand the section on how 'risky' are cousin marriages and inter-biraderi marriages. Specifically, since the definition of biraderi used in this paper emphasises the patriline, the authors might consider defining 'cousin' from a genetics viewpoint in order to clarify that all first cousins (mother's siblings children, and father's siblings children) are equivalent in terms of shared genetics.

I don't think this would go beyond the assumptions and findings of this research but there is anthropological evidence that in patrilineal societies, people may consider 'father's side' cousins to be genetically closer than those on the mother's side – an idea that does not match Mendelian genetics but that has led some families underestimating risks – e.g. by avoiding father's siblings children as partners for their children in favour of cousins on the mother's side, thinking these are less risky (see e.g. the chapter by Leila Prager, and the chapter by Shaw in *Cousin Marriages: between genetic risk and cultural change*, edited by Alison Shaw and Aviad Raz. Berghahn 2015. Also see Shaw's chapter in a book edited by Veronique Petit et al. *The Anthropological Demography of Health* OUP. 2020).

This is a very good point and we thank the reviewer for highlighting this. We added a sentence clarifying this concept in point 8 of the FAQ document . We also added the definition of cousin from a genetics viewpoint in the glossary.

The supplementary file will be useful for a general readership, and anyone – health professionals and lay people - already familiar with the UK (and wider) debates about ‘how risky’ cousin marriages are. The findings may also be of interest to social scientists including anthropologists (who these days often view kinship as entirely a social construct!), historians, and scholars of South Asia from all these disciplines.

We thank the reviewer for the positive feedback on the utility of the supplementary file.

I have therefore focused on clarifying points or terms used in the supplementary file that did not seem clear or consistent, most especially for non-specialists. These are listed below. [e.g. distinguish singular and plural meanings of biraderi - one biraderi /the biraderi system, vs. biraderis/biraderi groups, to distinguish consanguinity & consanguineous marriages, & check the use of of inter-biraderi & intra- biraderi]. Some of my suggestions implications for the main article as well, for consistency between the documents; these I indicate with a star *, and list briefly at the end.

Specific comments on the supplementary file

Questions 1-3

Lines:

5 – change ‘field’ to ‘fields’ (genetics, epidemiology, and anthropology are three separate fields - as far as I can see, only one of the research team is an anthropologist)

This has been changed in the text.

*10-11 – the term ‘population structure’ (in supplementary file & in main article), refers to genetic population structure, and ‘history’ refers to genetic history. However, for a lay audience these terms will have other meanings. Although there is a glossary, I suggest some rephrasing for clarity/accuracy.

12 – ‘do not translate well to individuals...’ -- perhaps rephrase: ‘may not be the same for people’ of other ethnicities.

This has been changed in the text.

*15 – 16 – ‘There is widespread interest in the history, dynamics and structure of populations’ – this is a general and ambiguous sentence that can be deleted. I suggest instead: ‘Research into a population’s genetic history, structure and dynamics can give information that can be compared with information about the populations’ social history, structure and dynamics.

We thank the reviewer for this suggestion. We have replaced the sentence accordingly.

*17-18 Also, a population’s unique social structure and cultural history – such as the biraderi system, and the practice of consanguineous marriage – can influence its genetic structure and genetic variability. (See the Glossary at the end of this paper for definitions of terms used).

We thank the reviewer for this suggestion. We have replaced the sentence accordingly.

Q 3-6

59 'In the DNA context' --- can these 4 words be deleted.

This has now been deleted.

*91 – consanguinity – in my understanding, this term does not refer to a practice - the practice of marrying someone who genetically related (as a second cousin or closer) - but to a quality of a relationship - technically this quality refers to the (quantifiable) extent to which two people have an ancestor in common.

So I think 'consanguinity' here should be changed to 'consanguineous marriage'

We thank the reviewer for the comment. This has been changed.

92-3 – endogamy – definition - I suggest you delete 'cultural' so it reads 'the practice of marrying...'
reasons: 'cultural' does add anything

The word 'cultural' has been deleted.

*105 'The Pakistani population' – does this mean, British Pakistanis? If so, say, the British Pakistani population is made up of people from different ethnic groups, such as...

We thank the reviewer for the comment. We meant the overall Pakistani population and not specifically the British Pakistani population.

105-6 'The society is organised into kinship networks passed through the male line....' .

British Pakistanis are not a 'society' but a population, and in Pakistan society there are other systems besides the biraderi system. And networks are not passed through the male line but biraderi identity is. So how about rephrasing as follows:

'Many British Pakistanis identify with a lineage-based system of kinship groups, which we refer to here as the biraderi system'

This has been changed accordingly.

*Regarding the use of the word biraderi, to avoid confusing your readers, I suggest that, because you are writing in English, you should follow English grammar - (even though in Urdu/Panjabi you would say 'one biraderi, two biraderi'). You can either add an 's' (not in italics) to the word when you are referring to two or more biraderis/ discussing biraderis in general, or else rephrase as 'biraderi groups'.

We agree with the reviewer on this point. We replaced throughout the manuscript 'biraderi' with 'biraderi groups' when referring to two or more biraderi groups.

108, please correct the grammar : - change 'one's' to 'their' and delete 'the' :
'... people tended to marry within their own biraderi, in order to reinforce hereditary social status, occupational identity and land ownership.

We thank the reviewer for spotting this error. We have changed it accordingly and deleted 'the'.

110-111 This next sentence can be expressed more simply:

To a certain extent, the biraderi system is still important in Pakistan and among British Pakistanis.

The sentence has been changed in the text.

*114 – change consanguinity to consanguineous

This has been changed in the text.

Q 7-11

164 – '(...such as a biraderi group)' - and you should italicise the word 'biraderi' for consistency.

We thank the reviewer for spotting this. According to editorial requirements we cannot use italics unless required for technical terms.

167 – in this line, biraderi should be plural to make sense in English. Please either add an 's' or change to biraderi groups.

This has been changed to biraderi groups.

168 – please change 'have' to 'result in' --- the marriage does not have the child, but the marriage may result in a child...

This has been changed accordingly.

169 – in this line, biraderi should be plural to make sense in English. Please either add an 's' or change to biraderi groups.

This has been changed to biraderi groups.

184-7 The sentence in brackets: there should be a plural form for the first to 'biraderi' in this sentence: '.....somewhat accurate for some biraderi groups, since most people who reported coming from those groups were more similar to others in their group than they were to people who reported coming from a different biraderi.

NB again, biraderi should be consistently in italics throughout

This has been changed to biraderi groups and italicised to be consistent throughout.

Pakistanis 188 – I suggest changing ‘the Bradford community’ to ‘Bradford Pakistanis’ (because although Bradford Pakistanis often refer to themselves in this way to distinguish themselves from other ethnic groups, the term implies a commonality and unity that is a bit misleading)

This has been changed accordingly.

202-4

Could you rephrase, to put more neutrally?,‘demonstrate the impact of marriage practices and partner choice on population structure and genomic diversity among Pakistanis’.

This has been changed accordingly.

210

Could you say, How were British Pakistanis involved in this study? (because although Pakistanis may refer to themselves in this way, ‘community’ implies a commonality and cultural unity that is somewhat misleading)

This has been changed accordingly.

213 can you change ‘and on the way findings can be shaped in ways that can improve...’ to ‘and on how research findings can be used to improve...’

This has been changed accordingly.

217...advice from people of Pakistani heritage living in Bradford who were....recognised locally as having a particular interest in and knowledge of their local communities, including the historic and contemporary importance of biraderi groups.

This has been replaced accordingly.

220 – could you change ‘from the Bradford Pakistani community’ to ‘born and raised in Bradford’ if this is indeed the case?

We thank the reviewer for the comment. However, Dr Sufyan Abid Dogra was born and raised in Pakistan.

Glossary – are the terms listed in order of appearance, or in alphabetic order?

They are listed in order of appearance.

*Population structure - This phrase occurs in the title. Could you change this to 'genetic population structure' -(please see below for my comment on demographic history, and on the title) because then the gloss is accurate.

This has been changed accordingly.

*Demographic history - This term did not come up in the supplementary file, it only occurs in the title of the main article.

We thank the reviewer for the comment. We rephrased the sentence in lines 15-16 to include 'genetic demographic history'. We also changed the entry in the glossary for clarity.

Demographic analysis is the statistical analysis of population characteristics such as population size, age, sex, ethnicity, in-migration and out-migration and demographic history refer to changes over time in these rates. Here you are referring to historical changes over time in the genetic structure of populations.

Do you need the word demographic? Could you simplify the title? It would also be useful for lay readers to have the word genetic in the title. A possible reworked title could be:

"Fine-scale genetic population structure and history of British Pakistanis"

We thank the reviewer for the comment. We believe the use of 'demographic history' is best suited in the context of our manuscript as it would better highlight the study of human populations characteristics in the past through quantitative analysis of genetic data.

*Consanguinity – see comment above about this word in line 91. If you use 'consanguineous marriage' instead, please add a gloss

*Consanguineous marriage – marriage to someone related genetically, as a second cousin or closer

We thank the reviewer for the comment. We replaced 'Consanguinity' with 'Consanguineous marriage' in the glossary.

Endogamy – the gloss is fine, but for the example in brackets, it would be helpful to add, 'or from the same biraderi'

This has been added accordingly.

*Biraderi - a kinship group with a shared identity (e.g. Jat, or Rajput) within a larger lineage-based system of kinship groups, which we refer to as the biraderi system.

This has been changed accordingly.

*Points for main article (page/line numbers give here may not be entirely accurate but worth checking the use of all these terms for consistency throughout the ms):

Population structure – genetic structure? Lines 38, 44, 48-9

We indicated more clearly genetic population structure throughout the FAQ document.

Change biraderi to the biraderi system, line 77; -- see above on singular/plural

We thank the reviewer for the comment. We replaced throughout the manuscript 'biraderi' with 'biraderi groups' when plural as stated before.

p. 18/ 562 change to particular biraderi groups

This has been changed accordingly.

P 3 line 101, also Line 587 – Pakistani community – or British Pakistanis?

This has been changed to British Pakistanis.

p. 2 line 84, add 'the ' – the Gupta empire

This has been added accordingly.

Section on broader impact:

line 584, and line 594 – inter-=within; intra= between; also p. 13?

We thank the reviewer for the comment. We clarify that we consider the prefix 'intra' for marriages within the same biraderi and 'inter' for marriages between individuals of different biraderi groups as stated previously. We added in brackets the word 'endogamous' for intra-biraderi marriages.

Line 589 – Bradford Pakistani community – or Bradford Pakistanis?

We thank the reviewer for the comment. We think Bradford Pakistani community is more appropriate in this context.